# NoiLIn: Improving Adversarial Training and Correcting Stereotype of Noisy Labels

**Jingfeng Zhang**[*]                                                    *jingfeng.zhang@riken.jp*
*RIKEN Center for Advanced Intelligence Project (AIP)*

**Xilie Xu**[*]                                                          *xuxilie@comp.nus.edu.sg*
*School of Computing, National University of Singapore*

**Bo Han**                                                              *bhanml@comp.hkbu.edu.hk*
*Department of Computer Science, Hong Kong Baptist University*

**Tongliang Liu**                                                       *tongliang.liu@sydney.edu.au*
*Trustworthy Machine Learning Lab, University of Sydney*

**Lizhen Cui**                                                          *clz@sdu.edu.cn*
*School of Software & Joint SDU-NTU Centre for Artificial Intelligence Research (C-FAIR), Shandong University*

**Gang Niu**                                                            *gang.niu@riken.jp*
*RIKEN Center for Advanced Intelligence Project (AIP)*

**Masashi Sugiyama**                                                    *sugi@k.u-tokyo.ac.jp*
*RIKEN Center for Advanced Intelligence Project (AIP)*
*Graduate School of Frontier Sciences, the University of Tokyo*

**Reviewed on OpenReview:** *https://openreview.net/forum?id=zlQXV7xtZs*

## Abstract

Adversarial training (AT) formulated as the minimax optimization problem can effectively enhance the model's robustness against adversarial attacks. The existing AT methods mainly focused on manipulating the inner maximization for generating quality adversarial variants or manipulating the outer minimization for designing effective learning objectives. However, empirical results of AT always exhibit the robustness at odds with accuracy and the existence of the cross-over mixture problem, which motivates us to study some label randomness for benefiting the AT. First, we thoroughly investigate noisy labels (NLs) injection into AT's inner maximization and outer minimization, respectively and obtain the observations on when NL injection benefits AT. Second, based on the observations, we propose a simple but effective method—NoiLIn that randomly injects NLs into training data at each training epoch and dynamically increases the NL injection rate once robust overfitting occurs. Empirically, NoiLIn can significantly mitigate the AT's undesirable issue of robust overfitting and even further improve the generalization of the state-of-the-art AT methods. Philosophically, NoiLIn sheds light on a new perspective of learning with NLs: NLs should not always be deemed detrimental, and even in the absence of NLs in the training set, we may consider injecting them deliberately. Codes are available in https://github.com/zjfheart/NoiLIn.

## 1 Introduction

Security-related areas require deep models to be robust against *adversarial attack* (Szegedy et al., 2014). To obtain *adversarial robustness*, *adversarial training* (AT) (Madry et al., 2018) would be currently the most

---

[*]Equal contribution

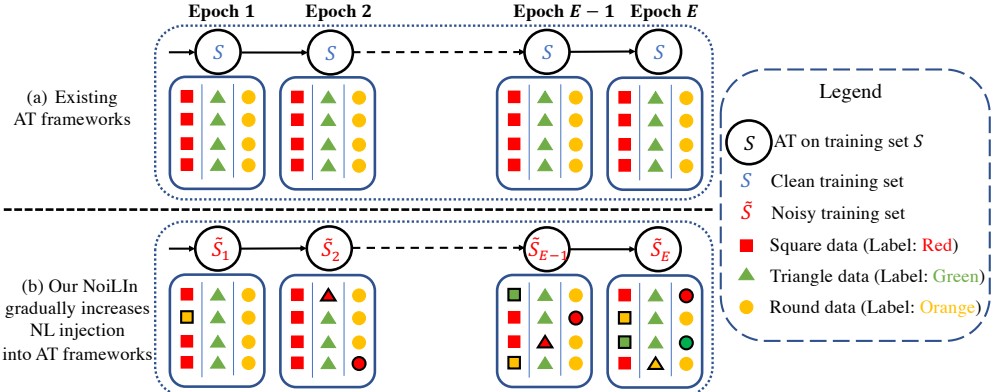

Figure 1: Comparisons between the existing AT framework (panel (a)) and our AT-NoiLIn framework (panel (b)). At each training epoch, AT-NoiLIn randomly flips labels of a portion of data (marked in the black edges), generating noisy-label training set $\tilde{S}$ for the learning.

effective defense that has so far not been comprehensively compromised (Athalye et al., 2018), in which AT is formulated as a minimax optimization problem with the *inner maximization* to generate adversarial data within small neighborhoods of their natural counterparts and the *outer minimization* to learn from the generated adversarial data.

However, AT empirically shows a severe tradeoff between the natural accuracy of natural test data and the robust accuracy of adversarial test data (Tsipras et al., 2019). Besides, Zhang et al. (2020b) showed AT has a *cross-over mixture problem*, where the adversarial variants of the natural data cross over the decision boundary and fall in the other-class areas. Even, Sanyal et al. (2021) showed the benchmark dataset (such as CIFAR (Krizhevsky, 2009)) contains some noisy data points. Furthermore. Donhauser et al. (2021) showed an unorthodox way to yield an estimator with a smaller robust risk, i.e., flipping the labels of a fixed fraction of the training data.

The above facts motivate us to explore manipulating labels for benefits that were largely overlooked by the existing AT studies. To enhance adversarial robustness further, the existing AT methods have mainly focused on manipulating the inner maximization or the outer minimization. For example, by manipulating the inner maximization for generating quality adversarial data, Zhang et al. (2020b) generated *friendly adversarial data* that are near the decision boundaries but are wrongly classified. On the other hand, by manipulating the outer minimization for designing the model's loss functions for the learning, Zhang et al. (2019b) designed TRADES that has two loss terms, i.e., the cross entropy loss on the natural data and the Kullback-Leibler (KL) divergence loss on the adversarial data. Since *NLs were often deemed to hurt the training* (Angluin & Laird, 1988), little efforts has been made to investigate noisy labels (NLs) in AT

To correct this negative stereotype of NLs, firstly, this paper thoroughly investigates NL injections in AT's every component. (*i*) We inject NLs in inner maximization. In each training minibatch, we randomly choose a portion of data whose adversarial variants are generated according to the flipped labels; the labels in outer minimization are intact for the learning. The label-flipped adversarial variants are far from the decision boundaries, thus the label-flipped adversarial data become no longer adversarial to the current model. In other words, label-flipped adversarial data serve more like natural data. As a result, when AT learns label-flipped adversarial data, the deep model's generalization gets increased but the robustness gets decreased.(*ii*) We inject NLs in outer minimization. We generate the adversarial data in each training minibatch according to the intact labels but randomly flip a portion of labels for the learning. NL injection in outer minimization leads to high *data diversity* for the learning, i.e., the deep model learns from different labels sets over the training process. From another perspective, NL injection in outer minimization implicitly averages deep models trained with different label sets. Therefore, NLs serve as regularization that can largely alleviate the AT's undesirable issue of robust overfitting (Rice et al., 2020) and even slightly enhances AT's best-epoch-checkpoint robustness. (*iii*) We inject NLs in both inner maximization and outer minimization. In each training epoch, we randomly choose a portion of data to flip their labels; then, we conduct standard

adversarial training (SAT (Madry et al., 2018)) on the noisy set. In other words, adversarial variants of this portion of data are generated according to the flipped labels, and their flipped labels are also used for the learning. The empirical results are very similar to case (*ii*).

Inspired by the above three observations, secondly, we propose a simple method "*NoiLIn*" that dynamically adjusts NL injection into AT (see Figure 1 for illustrations). In each training epoch, we first randomly flip a portion of labels of the training set, and then execute an AT method (such as SAT, TRADES (Zhang et al., 2019b), WAP (Wu et al., 2020)) using the noisy-label set. As the training progresses, we increase the flipping portion if there occurs robustness degradation (evaluated on a validation set).

Our contributions are summarized as follows. (a) We study NLs from a novel perspective, i.e., injecting NLs into the AT's training process; we observe that NL injection can even benefit AT. (b) Based on our observations, we propose to gradually increase the NL injection rate over the training process (i.e., NoiLIn). NoiLIn can significantly relieve the issue of robust overfitting of some AT methods such as SAT and TRADES. Using large-scale Wide ResNet Zagoruyko & Komodakis (2016) on the CIFAR-10 dateset Krizhevsky (2009), NoiLIn can even improve the generalization of current state-out-of-art (SOTA) TRADES-AWP Wu et al. (2020) by surprisingly 2.2% while maintaining its robustness (measured by auto attack (AA) Croce & Hein (2020)). (c) Philosophically, we should not always consider NLs to be detrimental. Even in the absence of NLs in the training set, we may consider injecting them deliberately.

## 2 Background and Related Work

There are many ways to improve the model's adversarial robustness, such as robustness certification (Hein & Andriushchenko, 2017; Weng et al., 2018; Wong & Kolter, 2018; Mirman et al., 2018; Tsuzuku et al., 2018; Lécuyer et al., 2019; Xiao et al., 2019; Cohen et al., 2019; Balunovic & Vechev, 2020; Zhang et al., 2020a; Singla & Feizi, 2020; Zhang et al., 2021a), Liptschiz regularization (Cisse et al., 2017; Moosavi-Dezfooli et al., 2019; Qin et al., 2019), incorporating attack modules (Yan et al., 2018), and detecting adversarial data (Metzen et al., 2017; Li & Li, 2017; Carlini & Wagner, 2017b; Tian et al., 2018; Ma et al., 2018; Lee et al., 2018; Pang et al., 2018; Smith & Gal, 2018; Roth et al., 2019; Liu et al., 2019; Yin & Rohde, 2020; Sperl et al., 2020; Cohen et al., 2020; Sheikholeslami et al., 2021; Chen et al., 2021a; Yang et al., 2020b; Qin et al., 2020; Tian et al., 2021; Wu et al., 2021b). Nevertheless, adversarial training (AT) still stands out as the most effective defense against the adaptive and strong attacks, attracting significant attention. This section briefly reviews the background of AT and provides related work.

**Adversarial Training (AT)** AT aims to a) classify the natural data $x$ correctly, and b) make the decision boundaries "thick" so that no data are encouraged to fall nearby the decision boundaries. AT's learning objective is as follows.

Let $(\mathcal{X}, L_\infty)$ be the input feature space $\mathcal{X}$ with the infinity distance metric $L_\infty(x, x') = \|x - x'\|_\infty$, and

$$\mathcal{B}_\epsilon[x] = \{x' \in \mathcal{X} \mid L_\infty(x, x') \leq \epsilon\} \tag{1}$$

be the closed ball of radius $\epsilon > 0$ centered at $x$ in $\mathcal{X}$. Given a dataset $S = \{(x_i, y_i)\}_{i=1}^n$, where $x_i \in \mathcal{X}$ and $y_i \in \mathcal{Y} = \{0, 1, \ldots, C-1\}$, AT's learning objective is formulated as a minimax optimization problem, i.e.,

$$\underbrace{\min_{f \in \mathcal{F}} \frac{1}{n} \sum_{i=1}^n \ell(f(\tilde{x}_i), y_i)}_{\text{outer minimization}}, \quad \underbrace{\tilde{x}_i = \arg\max_{\tilde{x} \in \mathcal{B}_\epsilon[x_i]} \ell(f(\tilde{x}), y_i)}_{\text{inner maximization}}. \tag{2}$$

Eq. (2) implies the AT's realization with one step of *inner maximization* on generating adversarial data $\tilde{x}_i$ and one step of *outer minimization* on fitting the model $f$ to the generated adversarial data $(\tilde{x}_i, y_i)$.

**Projected Gradient Descent (PGD)** To generate adversarial data, SAT Madry et al. (2018)[1] uses PGD to approximately solve the inner maximization. SAT formulates the problem of finding adversarial data as a

---

[1]Throughout this paper, we use AT to denote adversarial training methods in general and use SAT to specify the standard AT method by Madry et al. (2018).

constrained optimization problem. Namely, given a starting point $x^{(0)} \in \mathcal{X}$ and step size $\alpha > 0$, PGD works as follows:

$$x^{(t+1)} = \Pi_{\mathcal{B}[x^{(0)}]}\left(x^{(t)} + \alpha \operatorname{sign}(\nabla_{x^{(t)}} \ell(f_\theta(x^{(t)}), y))\right), \forall t \geq 0 \tag{3}$$

until a certain stopping criterion is satisfied. Commonly, the criterion can be a fixed number of iterations $K$. In Eq. equation 3, $\ell$ is the loss function in Eq. equation 2; $x^{(0)}$ refers to natural data or natural data corrupted by a small Gaussian or uniform random noise; $y$ is the corresponding true label; $x^{(t)}$ is adversarial data at step $t$; and $\Pi_{\mathcal{B}_\epsilon[x_0]}(\cdot)$ is the projection function that projects the adversarial data back into the $\epsilon$-ball centered at $x^{(0)}$ if necessary.

**Broad Studies on AT**   Existing literature studied/improved AT in many aspects such as customizing the inner maximization for simulating the better adversary (Goodfellow et al., 2015; Shafahi et al., 2019b; Zhang et al., 2019a; Wong et al., 2020; Vivek & Babu, 2020; Cai et al., 2018; Wang et al., 2019; Zhang et al., 2020b; Sriramanan et al., 2020) or the outer minimization for designing better loss functions/regularizer (Farnia et al., 2019; Song et al., 2019; Zhang et al., 2019b; Ding et al., 2020; Wang et al., 2020; Wu et al., 2020; Zhang et al., 2021b; Pang et al., 2021; Chen et al., 2021b), designing/searching robust network structures (Xie et al., 2020; Xie & Yuille, 2020; Guo et al., 2020; Sehwag et al., 2020; Yan et al., 2021; Du et al., 2021; Li et al., 2021; Huang et al., 2021; Li et al., 2021), employing multiple models (Pang et al., 2019; Yang et al., 2020a), augmenting the training data (Tramèr et al., 2018; Schmidt et al., 2018; Alayrac et al., 2019; Carmon et al., 2019; Najafi et al., 2019; Song et al., 2020; Lee et al., 2020; Gong et al., 2021; Gowal et al., 2021b; Wu et al., 2021a), and analyzing AT's intriguing properties (Tsipras et al., 2019; Ilyas et al., 2019; Stutz et al., 2019; Yin et al., 2019; Rice et al., 2020; Zhang & Zhu, 2019; Raghunathan et al., 2020; Yang et al., 2020c). Besides, recent studies showed AT could benefit other domains such as pre-training(Chen et al., 2020; Jiang et al., 2020), out-of-distribution generalization (Yi et al., 2021), inpainting (Khachaturov et al., 2021), interpretability (Ross & Doshi-Velez, 2018), fairness (Xu et al., 2021) and so on.

**Interaction between NLs and AT**   To avoid confusion, we clarify the differences between this paper and the existing studies of AT on NLs. NLs practically exist in the training set (Natarajan et al., 2013), and therefore, some studies have explored the effects of AT on NLs. Alayrac et al. (2019) showed that NLs degraded both generalization and robustness of AT, but robustness suffers less from NLs than generalization. Sanyal et al. (2021) showed robust training avoids memorization of NLs. Furthermore, Zhu et al. (2021) showed AT has a smoothing effect that can naturally mitigate the negative effect of NLs. Nevertheless, all those studies assumed NLs exist in the training set, which is detrimental to AT. In comparison, we assume the training set is noise-free. We treat NLs as friends and deliberately inject NLs to benefit AT in terms of relieving robust overfitting of SAT (Madry et al., 2018) and TRADES (Zhang et al., 2019b), even improving TRADES-AWP (Wu et al., 2020)'s generalization while maintaining its peak robustness.

**Relation with DisturbLabel (Xie et al., 2016)**   DisturbLabel, the most relevant work to NoiLIn, also randomly selects a small subset of data in each training epoch and then sets their ground-truth labels to be incorrect. Xie et al. (2016) studied only standard training (ST) for generalization. Differently, our work focuses on AT's robustness and generalization. Besides, compared with DisturbLabel that injects a small ratio NLs (20% or less) to alleviate ST's overfitting, AT needs injecting larger NLs ratio (e.g., 40% or more) to alleviate the AT's robust overfitting (see experiments in Figure 4). Compared with ST, AT encounters worsen situations of data overlaps of different classes, thus requiring stronger label randomness.

**Relation with Memorization in AT (Dong et al., 2022)**   The independent and concurrent work by Dong et al. (2022) explored various AT methods (i.e., training stability of SAT and TRADES methods) under completely random labels. Besides, Dong et al. (2022) proposed to mitigate SAT's and TRADES's issue of robust overfitting via adding temporal ensembling (TE) (Laine & Aila, 2017) as an additional learning objective to penalize the overconfident predictions. This shares a similar idea to injecting learned smoothing (Chen et al., 2021b) (i.e., adding several additional learning objectives) for mitigating the robust overfitting. We argue that our NoiLIn is simpler but is no worse than the above peer methods. TE and smoothness injection methods (Dong et al., 2022; Chen et al., 2021b) both need add additional learning objectives, which introduce more hyperparameters than NoiLIn's for finetuning the performance. Besides, without introducing additional learning objectives, NoiLIn saves computational resources.

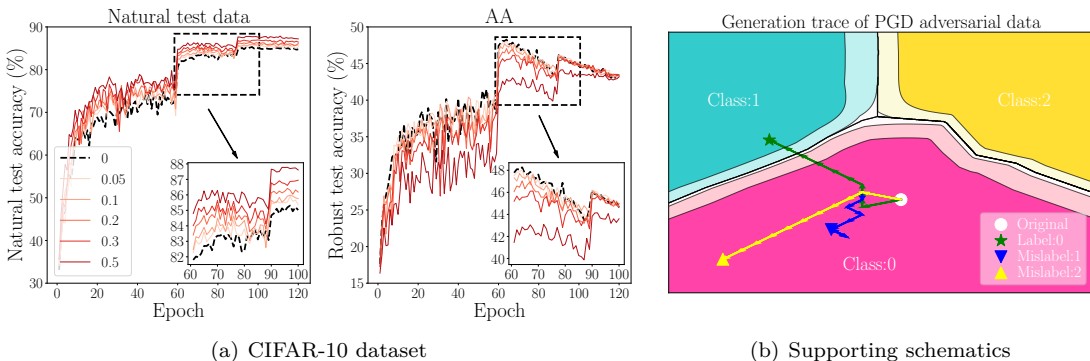

(a) CIFAR-10 dataset                    (b) Supporting schematics

Figure 2: Figure 2(a) shows the learning curves of injecting various levels of NL in inner maximization. (The number in the legend represents NL injection rate $\eta$; the black dash line represents SAT.) Figure 2(b) shows the generation trace of adversarial data on a synthetic ternary classification. (The white round denotes natural data. The blue trace is generated with correct label; both yellow and blue traces are generated with noisy label (NL). The color gradient represents the prediction confidence: the deeper color represents higher prediction confidence.) NL injection in inner maximization prevents generating effectively adversarial data.

**Relation with Label Manipulations Benefiting Robustness.** There are debates on whether label smoothing and logit squeezing (Shafahi et al., 2019a), and logit pairing (Engstrom et al., 2018; Mosbach et al., 2018) genuinely benefit adversarial robustness or mask gradients for overly reporting robustness. This paper avoids such a debate by evaluating our NoiLIn method using the strongest AutoAttack (AA) (Croce & Hein, 2020), which should thwart the concerns of gradient obfuscations (Athalye et al., 2018).

## 3 A Closer Look at NL injection in AT

Before introducing our algorithm—NoiLIn, we get started with a thorough understanding of NL injection in AT. Specifically, AT (see Eq. (2)) has two label-related parts: one is inner maximization for generating adversarial data; another is outer minimization for learning the generated adversarial data. In this section, we inject NLs in the two parts separately and obtain some intriguing observations.

### 3.1 NL Injection in Inner Maximization

**Experiment details** We conducted experiments of injecting NLs in inner maximization on the CIFAR-10 dataset Krizhevsky (2009). In Figure 2(a), we compare AT with NLs in inner maximization (red lines with different color degrees) and SAT (black dashed lines). We inject symmetric-flipping NLs Van Rooyen et al. (2015), where labels $y$ are flipped at random with the uniform distribution. In each training mini-batch, we randomly flip $\eta$ portion of labels of training data; then, the adversarial data are generated according to the flipped labels. The noise rate $\eta$ is chosen from $\{0, 0.05, 0.1, 0.2, 0.3, 0.5\}$. Note that it is exactly SAT when $\eta = 0$. The perturbation bound is set to $\epsilon_{\text{train}} = 8/255$; the number of PGD steps is set to $K = 10$, and the step size is set to $\alpha = 2/255$. The labels of outer minimization are intact for the learning. We trained ResNet-18 He et al. (2016) using stochastic gradient descent (SGD) with 0.9 momentum for 120 epochs with the initial learning rate of 0.1 divided by 10 at Epochs 60 and 90, respectively.

We evaluate the robust models based on the two evaluation metrics, i.e., *natural test accuracy* on natural test data and *robust test accuracy* on adversarial data generated by AutoAttack (AA) Croce & Hein (2020), respectively. The adversarial test data are bounded by $L_\infty$ perturbations with $\epsilon_{\text{test}} = 8/255$.

For completeness, in Appendix A, we demonstrate other attacks (such as PGD and CW attacks), and we also inject *pair-flipping* NLs Han et al. (2018), where labels $y$ are flipped between adjacent classes that are prone to be mislabeled.

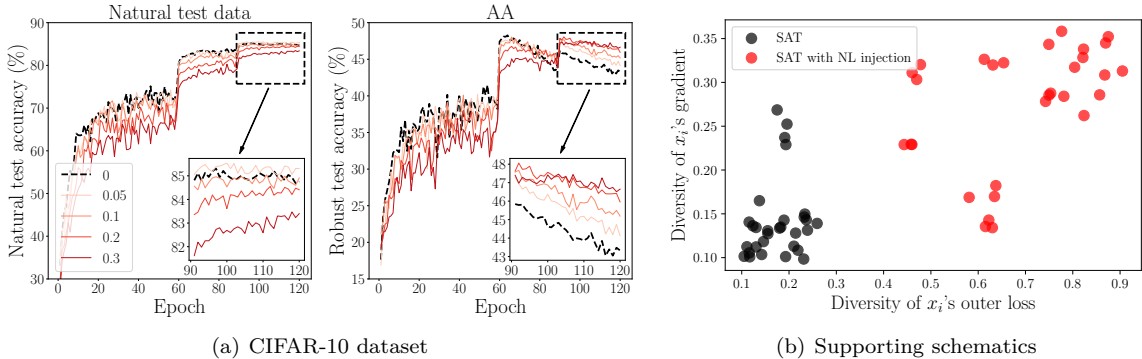

(a) CIFAR-10 dataset

(b) Supporting schematics

Figure 3: Figure 3(a) shows the learning curves of injecting various levels of NL in outer minimization. (The number in the legend represents NL injection rate $\eta$. The black dash line represents SAT.) Figure 3(b) shows data diversity comparisons between SAT (black dots) and SAT with NL injection (right dots). (Red dots are scattered at upper-right corner, which justifies that NL injection in outer minimization leads to high data diversity over the training epochs.) NL injection in outer minimization serves as a strong regularization that prevents robust overfitting.

**Observation (*i*)** *NL injection in inner maximization improves AT's generalization but degrades AT's robustness.* As shown in Figure 2(a), with the increasing of $\eta$, AT's generalization consistently increases (left panel), and AT's robustness consistently decreases (right panel).

Figure 2(b) illustrates the effects of NLs on generating adversarial data. We draw the generation traces of PGD adversarial data on a two-dimensional ternary classification dataset. We randomly choose a start point (white dot) whose correct label is "0" and plot the generation traces of adversarial data using correct label "0" (green trace), wrong label "1" (blue trace), and wrong label "2" (yellow trace), respectively. We find that NLs (wrong label "1" or "2") mislead the generation traces that should have pointed at the decision boundary. Instead, generation traces with NLs fall in the internal areas of class "0". Therefore, the generated NL-adversarial data are similar to the natural data of class "0". To further justify the above phenomenon, Figure 10 (in Appendix A) uses the CIFAR-10 dataset to corroborate that label-flipped adversarial data are similar to natural data.

Therefore, the reasons of Observation (*i*) are as follows. When PGD (see Eq. 3) generates adversarial data, NLs mislead PGD to find wrong directions to the decision boundary. Then, the label-flipped adversarial data becomes no longer adversarial to the current model, and they serve more like their natural counterparts. NL injection in inner maximization at each minibatch equals randomly replacing a part of adversarial data with natural data. Consequently, AT learning more natural data and less adversarial data leads to better generalization and worse robustness.

## 3.2 NL Injection in Outer Minimization

**Experiment details** Figure 3(a) shows the results of a series of experiments with various NL levels $\eta$ injected in outer minimization. Adversarial data are generated according to the intact labels. At every training minibatch, we randomly choose an $\eta$ portion of adversarial data to flip their labels for the learning. The other settings (e.g., the learning rate schedule, optimizer, network structures, dataset, $\epsilon_{\text{train}}$ and $\epsilon_{\text{test}}$, $K$, $\alpha$) are kept the same as Section 3.1. Note that we do not show $\eta > 0.3$ because injecting too much noise (especially at the early training stage) significantly destabilizes and even kills the training.

In Figure 3(a), we show the learning curves of injecting symmetric-flipping NLs Van Rooyen et al. (2015) . For completeness, in Appendix B, we show other attacks (such as the PGD and CW attacks) and also show pair-flipping NL Han et al. (2018) injection. In Appendix B, we find an intriguing phenomenon that pair-flipping NL injection obfuscates gradients of the CW attack. Specifically, pair-flipping NL injection significantly boosts the model's robustness against the CW attack.

**Observation** (*ii*)    *NL injection in outer minimization alleviates robust overfitting.* As shown in Figure 3(a), we find that at several last epochs, NL injection leads to consistently better robustness than SAT (black dashed line). Besides, the robustness at the final epoch improves with the increase of noise rate $\eta$. This shows that NL injection in outer minimization can significantly alleviate the undesirable issue of AT's robust overfitting Rice et al. (2020).

We interpret the Observation (*ii*) from data augmentation perspective. NL injection in outer minimization forces AT to learn the adversarial data $\tilde{x}$ with the flipped label $\tilde{y}$ occasionally. Considering an adversarial data point $(\tilde{x}, y)$ and its flipped label $\tilde{y}$, the objective loss function is $\ell(f(\tilde{x}), \tilde{y})$. Taking the MSE loss as an example, we can rewrite the loss $\ell(f(\tilde{x}), \tilde{y}) = ||f(\tilde{x}) - \tilde{y}||_2^2$ as $||(f(\tilde{x}) - \tilde{y} + y) - y||_2^2$, in which we view $\tilde{f}(\tilde{x}) := f(\tilde{x}) - \tilde{y} + y$ as noisy output. Through solving $\arg\min_{\tilde{x}_{\mathrm{aug}}} ||\tilde{f}(\tilde{x}) - f(\tilde{x}_{\mathrm{aug}})||$, we basically project noisy output onto input space and augment original adversarial data $\tilde{x}$ with $\tilde{x}_{\mathrm{aug}}$. Therefore, NL injection could increase the data diversity: at each training epoch, the model comes across different versions of data.

We manifest the above perspective through conducting a proof-of-concept experiment on the CIFAR-10 dataset. We compare SAT and SAT with NL injection. We randomly select 30 training data and collect the statistics of each data point at each epoch (120 epochs in total). Each data point $x_i$ corresponds to a 120-dimensional vector. $x_i$'s data diversity is reflected by the variance of its 120-dimensional vector.

Figure 3(b) illustrates the two statistics of each data point: 1) $x_i$'s *outer loss* (horizontal axis)—the loss of the generated adversarial data, i.e., $\ell(f(\tilde{x}_i), \tilde{y}_i)$ and 2) $x_i$'s *gradients* (vertical axis)—the $\ell_2$-norm of the weight gradient on $(\tilde{x}_i, \tilde{y}_i)$, i.e., $||\nabla_\theta \ell(f(\tilde{x}_i), \tilde{y}_i)||_2$. For notational simplicity, over the training process, $\tilde{y}_i$ is always equal to $y_i$ for SAT but not always equal to $y_i$ for SAT with NL injection (because $y_i$ is sometimes flipped to $\tilde{y}$).

Figure 3(b) shows that red rounds are scattered at the upper-right corner, and black squares are clustered at the lower-left corner, which justifies that SAT with NL injection has higher data diversity than SAT. Figure 3(b) echoes the empirical observations in Figure 3(a), which implies that the high data diversity may impede the the undesirable issue of robust overfitting.

**NL Injection in both Inner Maximization and Outer Minimization**

**Observation (iii)**    *NL injection in data has a similar effect as Observation (ii).* In each training minibatch, we randomly choose a portion of data to flip their labels. Then, we conduct SAT on them. We empirically find the performance is very similar to Observation (ii), with even a slightly better robustness. To avoid repetition and save space, we leave those results in Appendix C.

**Observation (iv)**    There exists another case where NL injection in inner maximization mismatches that in outer minimization. Specifically, at every minibatch, we choose a portion of data to flip labels in inner maximization and another portion of data to flip labels in outer minimization. In this setting, we observe slight degradation of robustness because label-flipped adversarial data are no longer serve the regularization purpose and are no longer adversarial as well to enhance robustness. The results can be found in Appendix D.

## 4    Method

The four observations in Section 3 give us some insights on designing our own methods: Obs. (i) and Obs. (iv) actually harms AT's robustness, but robustness is usually AT's main purpose. Obs. (ii) and Obs. (iii) have the similar effects that effectively mitigate robust overfitting. Inspired by the above Obs. (ii) or Obs. (iii), we propose our Algorithm 1 (i.e., NoiLIn) that dynamically increases NL injection rate once the robust overfitting occurs. The simple strategy can be incorporated into various effective AT methods such as SAT (Madry et al., 2018), TRADES (Zhang et al., 2019b), TRADES-AWP (Wu et al., 2020). As a result, NoiLIn can fix robust overfitting issues of SAT and TRADES and further enhance generalization of TRADES-AWP[2] (see experiments parts in Section 5.2).

---

[2]TRADES-AWP does not have robust overfitting issues because the AWP method has already fixed it. Besides, AWP has further enhanced TRADES's robustness, thus TRADES-AWP becomes the state-of-the-art AT method.

---

**Algorithm 1** NoiLIn: automatically increasing Noisy Labels Injection rate in AT methods

---

**Input:** network $f_\theta$, training set $S_{\text{train}}$, validation set $S_{\text{valid}}$, total epochs $E$, initial noise rate $\eta_{\min}$, maximal noise rate $\eta_{\max}$, sliding window size $\tau$, boosting rate $\gamma$
**Output:** adversarially robust network $f_\theta$
$\eta = \eta_{\min}$
**for** Epoch $e = 1, \ldots, E$ **do**
    Randomly flip $\eta$ portion of labels of training dataset $S_{\text{train}}$ to get $\tilde{S}$
    Update $f_\theta$ using $\tilde{S}$ by an AT method (such as SAT, TRADES, TRADES-AWP)
    Obtain robust validation accuracy $\mathcal{A}_e$ using $S_{\text{valid}}$
    **if** $\sum_{i=e-\tau}^{e} \mathcal{A}_i < \sum_{j=e-\tau-1}^{e-1} \mathcal{A}_j$ **then**
        $\eta = \min(\eta \cdot (1 + \gamma), \eta_{\max})$ // Boost NL injection rate $\eta$ if robust overfitting occurs.
    **end if**
**end for**

---

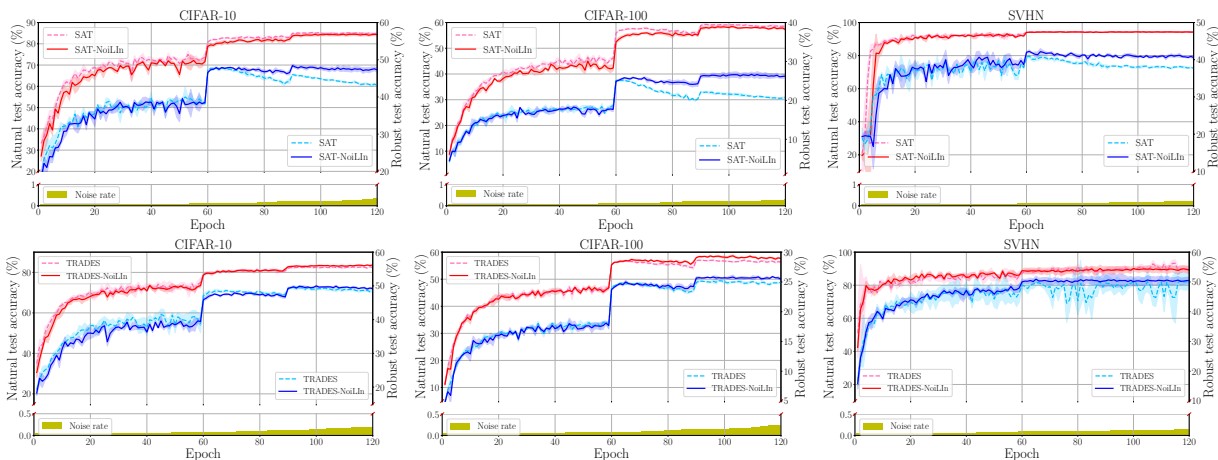

Figure 4: Evaluations on ResNet-18. We compare generalization (red lines) and robustness (blue lines, evaluated by the strongest AA attack) between our NoiLIn (solid lines) and two AT methods such as SAT and TRADES (dash lines) on three datasets—CIFAR-10, CIFAR-100, and SVHN. The yellow histogram below each figure reflects the change of NL injection rate over the training process. The shaded color reflects the standard deviations over the five repeated trails. To maintain figures' neatness, we put the training accuracy curves in Appendix D (Figure 17).

As stated in Algorithm 1, at the beginning of each epoch, we randomly flip $\eta$ portion of labels of the training dataset $S_{\text{train}}$ to get a noisy dataset $\tilde{S}$; then, we execute an existing AT method (e.g., SAT and TRADES) on the noisy dataset $\tilde{S}$. We monitor the training process using a clean validation set. We increase noise injection rate $\eta$ by a small ratio $\gamma$ once the robust overfitting occurs.

## 5 Experiment

### 5.1 NoiLin can Alleviate Robust Overfitting

Robust overfitting (Rice et al., 2020) is an undesirable phenomenon in some AT methods particularly such as SAT and TRADES, where after the first learning rate decay, the model's robustness stops increasing but begins to decrease.

In Figure 4, we incorporate our NoiLIn using symmetric-flipping NLs into two typical AT methods, i.e., SAT and TRADES. We adversarially train ResNet-18 on three datasets, i.e., CIFAR-10, CIFAR-100 and SVHN. We took 1000 data from training set for validation and took the remaining data for the training. The robust validation accuracy ($\mathcal{A}_e$ in Algorithm 1) was evaluated on PGD-10 attack for the consideration of speed. For the hyperparameter setting of AT-NoiLIn, the noise rate $\eta$ was initialized as $\eta_{\min} = 0.05$ with

Table 1: Evaluations on Wide ResNet. We reported the test accuracy of the best checkpoint and that of the last checkpoint as well as the gap between them—"best±std. /last±std. (gap)".

| Defense | Natural | C&W$_\infty$-100 | AA |
|---|---|---|---|
| SAT (Madry et al., 2018) | **87.00**±0.556/**87.13**±0.167 (+**0.13**) | 53.47±0.174/46.75±0.272 (−6.72) | 51.06±0.471/45.29±0.247 (−5.77) |
| SAT+TE (Dong et al., 2022) | 85.84±0.155/86.25±0.114 (+0.41) | 55.91±0.128/53.23±0.154 (−2.68) | 52.21±0.262/50.32±0.243 (−1.89) |
| SAT-NoiLIn | 85.92±0.400/86.43±0.343 (+0.51) | **55.99**±0.352/**53.79**±0.579 (−**2.20**) | **52.50**±0.278/**50.62**±0.555 (−**1.88**) |
| TRADES (Zhang et al., 2019b) | 84.92±0.221/85.16±0.062 (+0.25) | 53.69±0.054/49.89±0.121 (−3.80) | 52.54±0.313/47.95±0.419 (−4.59) |
| TRADES+TE (Dong et al., 2022) | 85.25±0.265/85.78±0.222 (+0.53) | 53.30±0.308/51.42±0.092 (−1.88) | 52.56±0.401/50.09±0.341 (−2.47) |
| TRADES-NoiLIn | **84.39**±0.142/**85.89**±0.076 (+**1.50**) | **54.37**±0.246/**51.69**±0.201 (−**2.68**) | **53.14**±0.326/**50.16**±0.050 (−**2.88**) |
| TRADES-AWP (Wu et al., 2020) | 84.48±0.377/84.96±0.040 (+0.48) | 58.88±0.087/57.60±0.194 (−1.28) | 55.88±0.182/54.91±0.207 (−0.97) |
| TRADES-AWP-NoiLIn | **86.69**±0.153/**87.13**±0.372 (+**0.44**) | **59.98**±0.319/**59.54**±0.348 (−**0.44**) | **56.12**±0.139/**55.89**±0.324 (−**0.23**) |

$\eta_{\max} = 0.6$, $\tau = 10$ and $\gamma = 0.1$. For the hyperparameter setting of TRADES-NoiLIn, we set $\eta_{\min} = 0.05$, $\eta_{\max} = 0.4$, $\tau = 10$ and $\gamma = 0.05$. We trained ResNet-18 using SGD with 0.9 momentum for 120 epochs with the initial learning rate of 0.1 (on CIFAR datasets and 0.01 on SVHN) and divided by 10 at Epochs 60 and 90, respectively. We compare the generalization evaluated on natural test data (red lines) and robustness evaluated on AA adversarial data (Croce & Hein, 2020) (blue lines) between typical AT methods (such as SAT and TRADES; dash lines) and NoiLIn counterparts (such as SAT-NoiLIn and TRADES-NoiLIn; solod lines). Besides, at the bottom of each panel, we also demonstrate the dynamic schedule of noise rate $\eta$ using the yellow histogram.

Figure 4 empirically validates the efficacy of NoiLIn to alleviate robust overfitting. When the learning rate decays, the robust test accuracy of NoiLIn keeps steady and even rises slightly instead of immediately dropping. Besides, we observe that when the robust overfitting occurs, the noise rate (yellow pillars) gradually rises to deliver a stronger regularization to combat robust overfitting.

## 5.2 Performance Evaluations on Wide ResNet

To manifest the power of NoiLIn, we adversarially train Wide ResNet (Zagoruyko & Komodakis, 2016) on CIFAR-10 by incorporating our NoiLIn strategy using symmetric-flipping NLs to three common and effective AT methods, i.e., SAT, TRADES and TRADES-AWP (adversarial weight perturbation). In Table 1, we used WRN-32-10 for AT-NoiLIn that keeps same as Madry et al. (2018) and used WRN-34-10 for TRADES-NoiLIn and TRADES-AWP-NoiLIn that keeps same as Zhang et al. (2019b); Wu et al. (2020). The hyperparameters of AWP exactly follows Wu et al. (2020). Other training settings (e.g., learning rate, optimizer, the hyperparameters for scheduling noise rate) keeps the same as Section 5.1. In Table 1, we also compare independent and contemporary work, i.e.,"AT+ TE (temporal ensembing)" (Dong et al., 2022).

On the current SOTA method—TRADES-AWP, our NoiLIn could further enhance its natural test accuracy by surprisingly 2% while maintaining (even slightly improving) its robustness. Besides, on natural test accuracy, NoiLIn seems to have a bigger negative impact on SAT than TRADES. In terms of learning objectives, SAT has one cross-entropy loss on adversarial data; TRADES has two loss, i.e., cross-entropy loss on natural data + 6×KL divergence loss between natural data and adversarial data, in which KL loss does not involve labels and only cross-entropy loss involves labels. Therefore, NoiLIn has less negative impact on TRADES than AT on natural accuracy.

## 5.3 Ablation Study

In this section, we conduct ablation studies. We show NoiLIn is more effective in larger perturbation bounds $\epsilon$, is less sensitive to weight decay. Besides, we compare NoiLIn with label smoothing (LS), and conduct extensive ablation studies such as NoiLIn with various learning rate schedulers and various noisy-label training set.

**NoiLIn under larger $\epsilon$.** NoiLIn is more effective for larger $\epsilon$. Robust overfitting becomes even worse when $\epsilon$ becomes larger. We compare SAT and SAT-NoiLIn under larger $\epsilon$ using ResNet-18 on CIFAR-10 dataset. The training settings of SAT and SAT-NoiLIn keep same as Section 5.1 except $\epsilon$. We choose $\epsilon_{\text{train}}$ from {8/255,10/255,12/255,14/255,16/255}. The step size $\alpha$ for PGD was $\epsilon/4$. Robust test accuracy was evaluated on adversarial data bounded by $L_\infty$ perturbations with $\epsilon_{\text{test}} = \epsilon_{\text{train}}$.

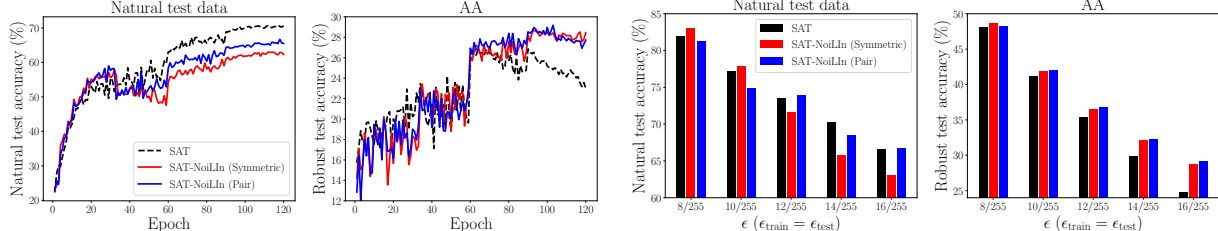

(a) Robust overfitting is more severe under larger $\epsilon = 16/255$.  (b) NoiLIn enhances robustness further under larger $\epsilon$.

Figure 5: Figure 5(a) shows the learning curves of SAT-NoiLIn under $\epsilon_{\text{train}} = 16/255$. (Robust test accuracy was evaluated on AA attacks with $\epsilon_{\text{test}} = 16/255$). Figure 5(b) shows comparisons between SAT and SAT-NoiLIn under larger $\epsilon_{\text{train}}$. (We reported the test accuracy evaluated at the best checkpoint according to the best robust accuracy. All figures keep $\epsilon_{\text{train}} = \epsilon_{\text{test}}$.) We abbreviate SAT-NoiLIn using symmetric-flipping NLs as "SAT-NoiLIn (Symmetric)" and SAT-NoiLIn using pair-flipping NLs as "SAT-NoiLIn (Pair)".

Figure 5(a) illustrates the learning curves of SAT and SAT-NoiLIn under $\epsilon_{\text{train}} = \epsilon_{\text{test}} = 16/255$. Figure 5(b) reports performance of the best checkpoints under different $\epsilon$. The best-checkpoints are chosen according to the best robust accuracy. Under larger $\epsilon$, SAT has more severe issue of robust overfitting and the robustness becomes lower. It is because AT fit data's neighborhoods, and the neighborhood becomes exponentially larger w.r.t. data's dimension even if $\epsilon$ become slightly larger. This will cause larger overlap issues between classes, which requires larger label randomness for the learning. Fortunately, under larger $\epsilon$, NoiLIn is more effective in relieving robust overfitting and even further improves adversarial robustness.

**NoiLIn's sensitivity to weight decay.** Pang et al. (2021) empirically found different values of weight decay can largely affect AT's adversarial robustness. In the ablation study, we explore the sensitivity of NoiLIn to weight decay. Figure 6 shows NoiLIn can reduce AT methods' sensitivity on weight decay.

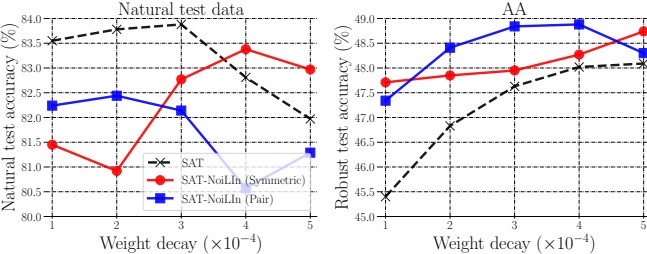

Figure 6: Test accuracy of robust models trained under different values of weight decay.

In Figure 6, we conduct Algorithm 1 under different values of weight decay and compared the SAT-NoiLIn with SAT. We use ResNet-18 and evaluate robustness at the best checkpoint. The settings keep the same as Pang et al. (2021). From the right panel of Figure 6, both red and blue lines are higher and flatter than black lines across different values of weight decay. It signifies that noise injection can enhance adversarial robustness of SAT and meanwhile make SAT less sensitive to varying values of weight decay.

**NoiLIn's relation with label smoothing (LS).** When taking the expectation w.r.t. all training epochs, NL injection is similar to employing the technique of label smoothing Szegedy et al. (2016); Müller et al. (2019) into AT (AT-LS) (Cheng et al., 2020; Pang et al., 2021). Despite the mathematical similarity, we empirically found that NoiLIn can relieve robust overfitting while LS cannot.

In Table 2, we reported robustness evaluations of AT with LS and AT-NoiLIn using ResNet-18 on CIFAR-10 dataset and set the training settings same as Section 5.1. AT-LS uses the $\rho$ level of smoothed label $\bar{y}$ in outer minimization while using the one-hot label $y$ in inner maximization. The $\rho$ level of smoothed label $\bar{y}$ is a $C$-dimensional vector with the $y$-th dimension being $1 - \rho$ and all others being $\frac{\rho}{C-1}$. We implement LS with $\rho = 0.1$ only in outer minimization ("AT-LS-outer") and LS with $\rho = 0.1$ in both inner maximization and outer minimization ("AT-LS-both"). Further, we even conduct LS in both inner maximization and

Table 2: Comparisons between SAT, AT-LS-outer ($\rho = 0.1$), AT-LS-both ($\rho = 0.1$), AT-LS-adaptive and SAT-NoiLIn (Symmetric). We report the test accuracy of the best checkpoint and that of the last checkpoint as well as the gap between them—"best/last (gap)".

| Defense | Natural | C&W$_\infty$-100 | AA |
|---|---|---|---|
| SAT Madry et al. (2018) | 81.97/84.76 (+2.79) | 49.53/45.12 (−4.71) | 48.09/43.30 (−4.79) |
| AT-LS-outer Pang et al. (2021) | 82.76/85.15 (+2.39) | 50.06/45.40 (−4.66) | 48.60/43.44 (−5.16) |
| AT-LS-both | 82.28/84.34 (+2.06) | 49.79/46.17 (−3.62) | 48.44/44.59 (−3.85) |
| AT-LS-adaptive | **82.68/85.20 (+2.52)** | 50.03/46.72 (−3.31) | 48.68/45.23 (−3.45) |
| SAT-NoiLIn | 82.97/83.86 (+0.89) | **51.21/50.36 (−0.85)** | **48.74/47.93 (−0.81)** |

outer minimization with dynamic LS level $\rho$ ("AT-LS-dynamic"). The schedule of $\rho$ is set exactly same as that of $\eta$ in AT-NoiLIn. We report the test accuracy at the best/last epoch and the gap between them in Table 2. Extensive comparisons between AT-LS-outer under other different $\rho$ and AT-NoiLIn are reported in Appendix D.1.

We found none of LS strategies can effectively relieve the robust overfitting. Table 2 shows that compared with variants of AT-LS, AT-NoiLIn can largely relieve robust overfitting, while LS fails to make any effect. The reasons may be that AT-LS uses fixed smoothed labels, which is similar to AT using fixed one-hot labels, thereby having lower data diversity during the training process; by contrast, NoiLIn randomly flips labels at each epoch, leading to higher data diversity and thus effectively preventing robust overfitting (see Section 3.2).

In addition, compared with LS, NoiLIn saves $(C−1)$ log operations for each data. For each adversarial training data $\tilde{x}$ whose prediction is $p(\tilde{x})$, the calculation of the cross-entropy loss of AT-LS is $-\sum_{j=1}^{C} \bar{y}_j \log p_j(\tilde{x})$ while that of SAT-NoiLIn is $-\log p_{\tilde{y}}(\tilde{x})$.

**NoiLIn under different learning rate (LR) schedulers.** Rice et al. (2020) showed robust overfitting ubiquitously occurs in AT under different LR schedulers. We conducted SAT-NoiLIn under different LR schedulers using ResNet-18 on CIFAR-10 dataset and kept all the training settings, such as the optimizer and hyperparameters of noise rate, exactly same as Section 5.1 except the LR scheduler. We report the

Table 3: Different learning rate schedule

| Defense | LR scheduler | Natural | C&W$_\infty$-100 | AA |
|---|---|---|---|---|
| SAT Madry et al. (2018) | piecewise | 81.97/84.76 (+2.79) | 49.53/45.12 (−4.71) | 48.09/43.30 (−4.79) |
| | piecewise | 82.97/83.86 (+0.89) | 51.21/50.36 (−0.85) | 48.74/47.93 (−0.81) |
| SAT-NoiLIn | multiple decay | 81.74/83.35 (+1.61) | 49.42/48.59 (−0.83) | 47.50/46.22 (−1.28) |
| (Symmetric) | cosine | **84.86/85.02 (+0.16)** | 50.83/50.59 (−0.24) | 48.24/48.08 (−0.16) |
| | cyclic | 83.48/83.48 (+0.00) | **51.41/51.41 (+0.00)** | **49.05/49.05 (+0.00)** |

test accuracy of the best checkpoint and that of the last checkpoint as well as the gap between them in Table 3 and show the learning rate schduler in Figure 18. We observe the gap between test accuracy of the best checkpoint and that of the last checkpoint largely narrows with automatic NL injection, which clearly indicates NoiLIn can mitigate robust overfitting under all different LR schedulers. Further, we demonstrate the learning curves of SAT-NoiLIn under various LR schedulers in Figure 18 (Appendix D.2), which again validates NL injection can relieve overfitting.

**NoiLIn under noisy-label training set.** Readers may feel curious about what happens when NoiLIn meets noisy-label set because training set is often noisy-labeled Xiao et al. (2015). We compare AT-NoiLIn and SAT on noisy CIFAR-10 dataset using ResNet-18. We use symmetric-flipping NL to construct noisy CIFAR-10. The fraction of NL is chosen from $\{0\%, 5\%, 10\%, 20\%, 30\%, 50\%\}$. Note that when the fraction of NL is 0%, the training set is exactly the clean CIFAR-10 dataset. The training settings of SAT and AT-NoiLIn kept same as Section 5.1. In Figure 7, we show the performance of SAT (black lines) and AT-NoiLIn (red lines) on noisy-label training set that contains different fractions of NL.

We find red lines are always above black lines, which indicates that even with noisy-label training set, our NoiLIn is still an effective method to improve the AT methods.

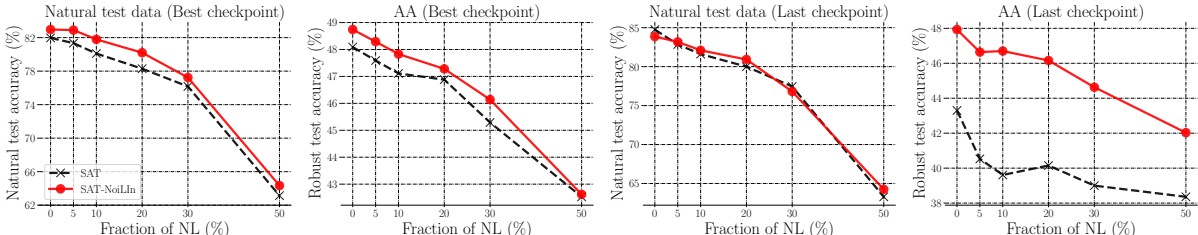

Figure 7: Comparisons between SAT and SAT-NoiLIn on noisy-label training set containing different fractions of NL.

**NoiLIn with extra unlabeled training data**    Carmon et al. (2019) and Gowal et al. (2021a) showed that additional unlabeled data can enhance adversarial robustness. Therefore, we conduct NoiLIn with additional training unlabeled data provided by Carmon et al. (2019). We train WRN-28-10 with the cosine decay learning rate schedule. We set the ratio of labeled-to-unlabelled data per batch to 3:7 following Gowal et al. (2021a). The total training epoch is 200 and the batch size is 256. Other hyper-parameters for injecting label noise are the same as Section 5. In Table 4, we compare the natural test accuracy and robust test accuracy (evaluated by AA attack) between NoiLIn and Robust Self-training (Carmon et al., 2019). Note that the results of Robust Self-training with extra data Carmon et al. (2019) are copied from AutoAttack leaderboard (Croce et al., 2020). Surprisingly, Table 4 shows that NoiLIn can enhance both natural and robust test accuracy, which validates the superiority of the NoiLIn method.

Table 4: Evaluations of NoiLIn with additional unlabeled data on WRN-28-10.

| Defense | Natural | AA |
| --- | --- | --- |
| Robust Self-teraining by Carmon et al. (2019) | 89.69 | 59.53 |
| NoiLIn with additional unlabeled data | **91.11** | **59.91** |

## 6  Conclusion

In this paper, we claim that NL injection can benefit AT. We have explored the positive effects of NL injection in inner maximization and in outer minimization, respectively. Our observations have motivated us to propose a simple but effective strategy, namely "NoiLIn", to combat the issue of robust overfitting and even enhance robustness further (e.g., especially under larger $\epsilon$).

There are two limitations of our current work. 1) We empirically verified the benefits of NoiLIn, but it is still difficult to explicitly, not intuitively, explain why NL injection improves robustness. 2) We do not know the optimal rate of NL injection at each training epoch, except leveraging validation set. In the future, we will attempt to address these limitations.

### Acknowledgment

JZ was supported by JST Strategic Basic Research Programs, ACT-X, Grant No. JPMJAX21AF and JSPS Grants-in-Aid for Scientific Research (KAKENHI), Early-Career Scientists, Grant No. 22K17955, Japan. BH was supported by the RGC Early Career Scheme No. 22200720, NSFC Young Scientists Fund No. 62006202, Guangdong Basic and Applied Basic Research Foundation No. 2022A1515011652, RIKEN Collaborative Research Fund and HKBU CSD Departmental Incentive Grant. LTL was partially supported by the Australian Research Council Project DP180103424, DE190101473, IC190100031, DP220102121. LC was supported by the National Key R&D Program of China No. 2021YFF0900800, NSFC No.91846205, SDNSFC No.ZR2019LZH008, Shandong Provincial Key Research and Development Program (Major Scientific and Technological Innovation Project) No.2021CXGC010108. MS was supported by JST AIP Acceleration Research Grant Number JP- MJCR20U3 and the Institute for AI and Beyond, UTokyo

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

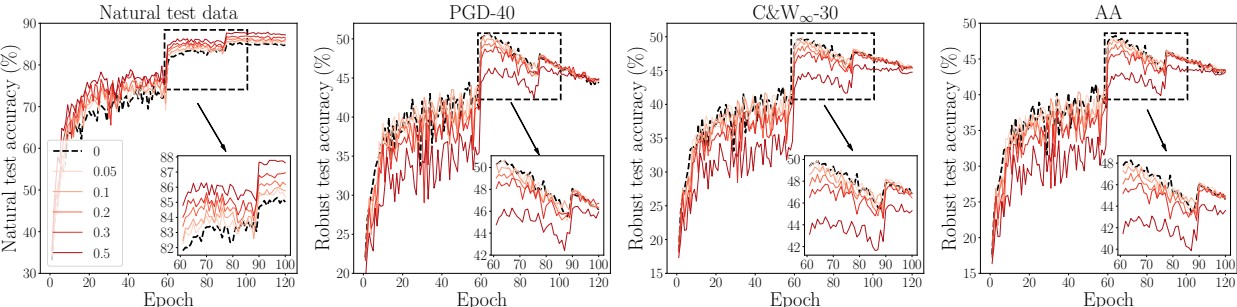

Figure 8: The learning curves of injecting various levels of symmetric-flipping NL in inner maximization. The number in the legend represents noise rate $\eta$.

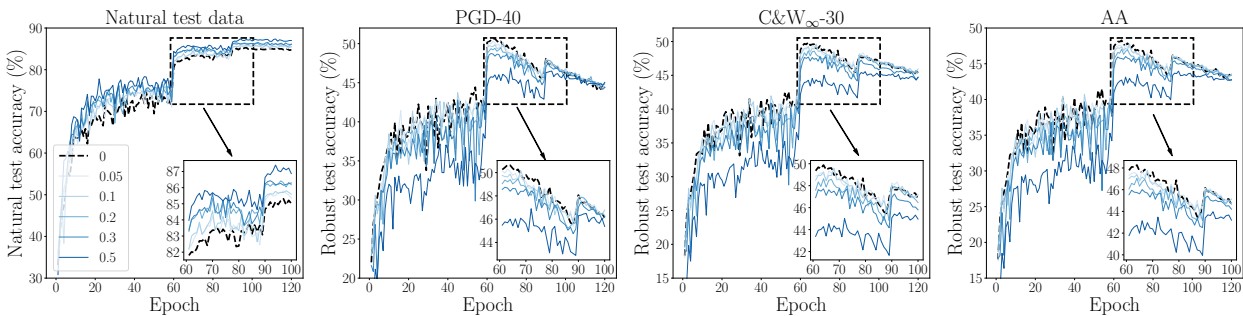

Figure 9: The learning curves of injecting various levels of pair-flipping NL in inner maximization. The number in the legend represents noise rate $\eta$.

## A    NL Injection in Inner Maximization

Figure 8 is symmetric-flipping NL injection in inner maximization; Figure 9 is pair-flipping NL injection in inner maximization. We evaluated the robust models based on the four evaluation metrics, i.e., natural test accuracy on natural test data, robust test accuracy on adversarial data generated by PGD-40 Carmon et al. (2019), C&W$_\infty$-30 ($L_\infty$ version of C&W Carlini & Wagner (2017a) optimized by PGD-30), and AutoAttack Croce & Hein (2020) (AA), respectively. The adversarial test data are bounded by $L_\infty$ perturbations with $\epsilon_{\text{test}} = 8/255$. Figures 8 and 9 consistently show NL injection in inner maximization improves AT's generalization and degrades AT's robustness. To further justify the above phenomenon in Figure 2(b), in Figure 10, we used a real-world dataset—CIFAR-10—to reveal the similarity among "PGD with NL" (PGD adversarial data with wrong labels), "PGD with CL" (PGD adversarial data with correct labels), and "Natural" (natural data). We conducted a standard adversarial training for 120 epochs and saved the model's checkpoint at every epoch. At every checkpoint, we randomly selected 1000 training data and generated "PGD with NL" and "PGD with CL" of these data. We used model's Kullback–Leibler (KL) loss Zhang et al. (2019b) as the similarity metric (the smaller value means larger similarity). In the left panel of Figure 10, we compared similarity among "PGD with NL", "PGD with CL", and "Natural". Besides, in the right panel of Figure 10, we used the correct labels to calculate the cross-entropy loss of "PGD with NL", "PGD with CL", and "Natural", respectively.

The left panel of Figure 10 shows that the value of KL(PGD with NL, Natural) (red solid line) is apparently lower than that of KL(PGD with CL, Natural) (blue solid line). Besides, the right panel of Figure 10 shows that the cross-entropy loss of "PGD with NL" is almost identical to that of "Natural". Compared with "PGD with CL", "PGD with NL" is more similar to "Natural" (natural data). Therefore, label-flipped adversarial data are similar to natural data.

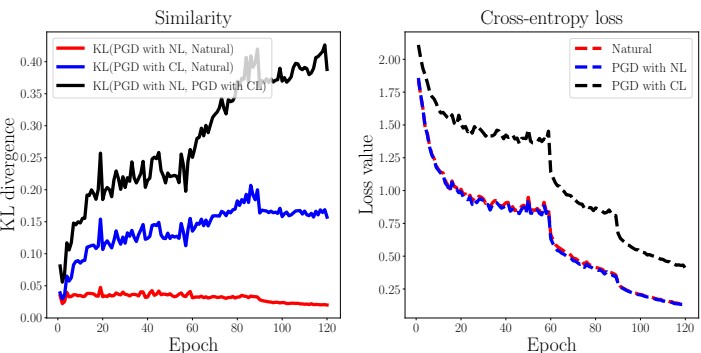

Figure 10: The similarity among PGD with CL (adversarial data generated with correct label), PGD with symmetric-flipping NL (adversarial data generated with noisy label), and Natural data on CIFAR-10 dataset.

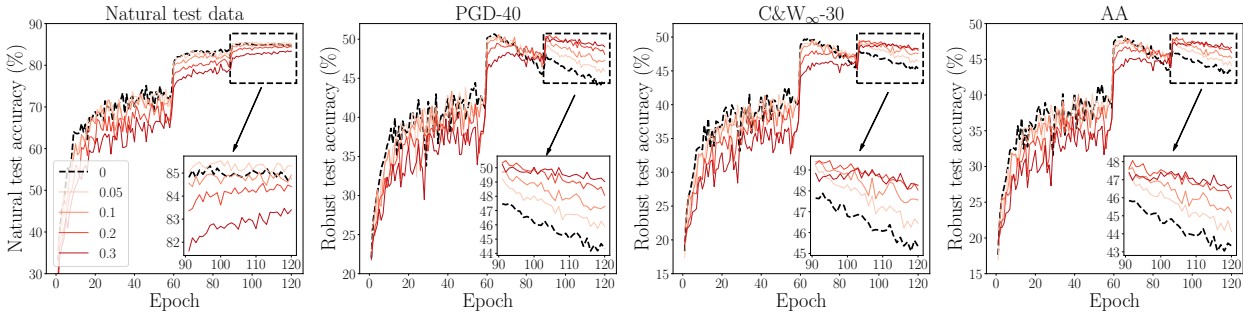

Figure 11: The learning curves of injecting various levels of symmetric-flipping NL in outer minimization. The number in the legend represents noise rate $\eta$

## B NL Injection in Outer Minimization

Figure 11 is symmetric-flipping NL injection in outer minimization; Figure 12 is pair-flipping NL injection in outer minimization. The evaluation metrics keep the same as Appendix A.

From the third panel of Figure 12, we observe that NL in outer minimization can significantly improve the model's robustness against the C&W attack, especially injecting pair-flipping label noise.

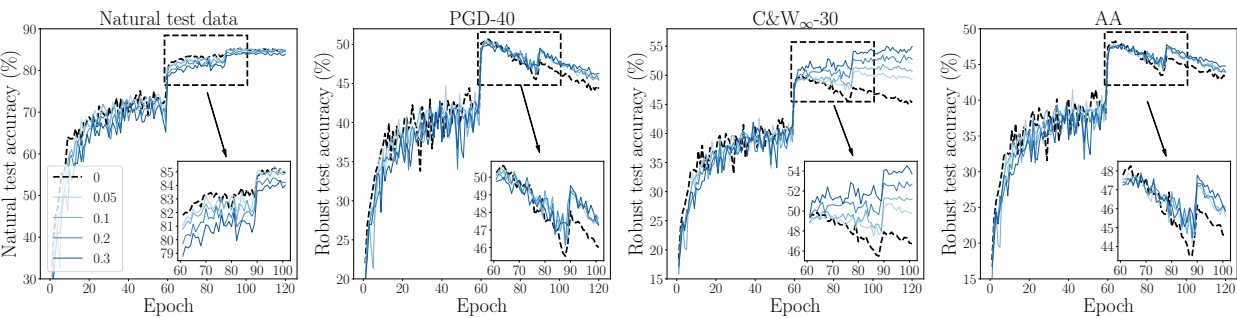

Figure 12: The learning curves of injecting various levels of pair-flipping NL in outer minimization. The number in the legend represents noise rate $\eta$

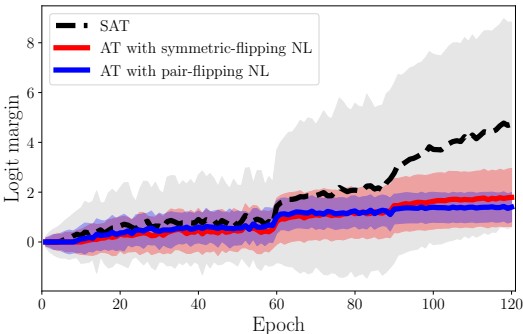

Figure 13: We compare the median *logit margin* of natural test data and its standard deviation over the training process of SAT (black dashed line), that of AT with symmetric-flipping NL in outer minimization (red line), and that of AT with pair-flipping NL in outer minimization (blue line). We find NL in outer minimization reduces logit margins, thus obfuscating gradients of the C&W attack Carlini & Wagner (2017a).

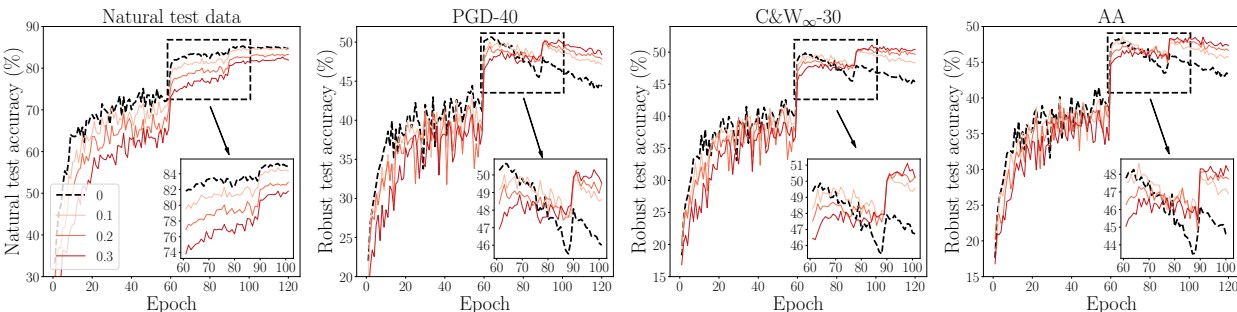

Figure 14: The learning curves of injecting symmetric-flipping NL into both inner maximization and outer minimization. The number in the legend represents noise rate $\eta$.

To figure out the reason for this interesting phenomenon, at every training epoch, we illustrated the median *logit margin* $\left(f_\theta^y(x) - \max_{j \neq y} f_\theta^j(x)\right)$ over all natural test data and its standard deviation in Figure 13. We compare SAT (black dashed line) with AT with NL (the red line for injecting symmetric noise and the blue line for injecting pair flipping noise). As shown in Figure 13, injecting NL in outer minimization leads to a small logit margin, especially injecting pair-flipping NL (the blue line).

Note that the implementation of the C&W attack follows

$$\tilde{x} = \arg\max_{\tilde{x} \in \mathcal{B}_\epsilon[x]} \left( \max(\max_{j \neq y} f_\theta^j(\tilde{x}) - f_\theta^y(\tilde{x}) - \kappa, 0) \right), \tag{4}$$

where $\kappa$ is a positive constant value that aids the optimization. When the logit margin is small and close to 0, the starting point of optimization of the inner loss of the C&W attack (i.e., $(\max(\max_{j \neq y} f_\theta^j(\tilde{x}) - f_\theta^y(\tilde{x}) - \kappa, 0))$ in Eq. equation 4) is also small and close to 0. This will incur the gradient vanishing problem and hurdle the optimization for finding the C&W-adversarial data. Therefore, NL in outer minimization obfuscates gradients of C&W attacks by reducing logit margins.

# C   Injecting NL in Both Inner Maximization and Outer Minimization

**Observation (iii)**   We report the learning curves of ResNet-18 trained by AT with NL injection in both inner maximization and outer minimization on CIFAR-10 dataset in Figure 14 (symmetric-flipping NL) and Figure 15 (pair-flipping NL). The noise rate $\eta$ is chosen from $\{0.1, 0.2, 0.3\}$. The detailed training settings (e.g., the optimizer and learning schedule) are the same as Section 3.1.

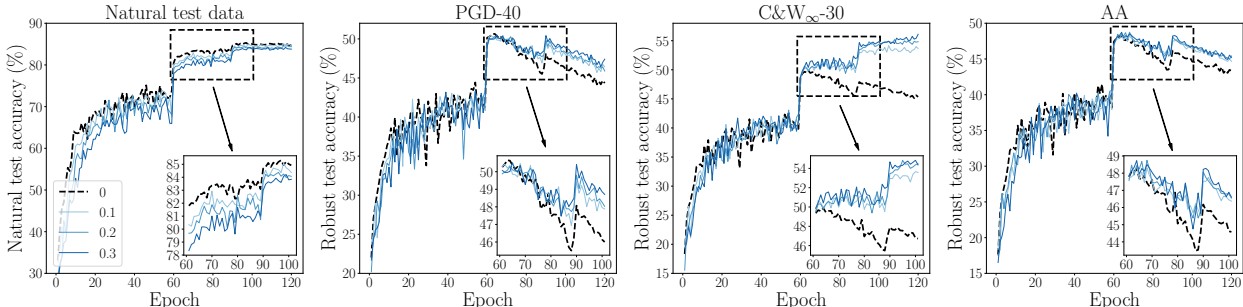

Figure 15: The learning curves of injecting pair-flipping NL into both inner maximization and outer minimization. The number in the legend represents noise rate $\eta$.

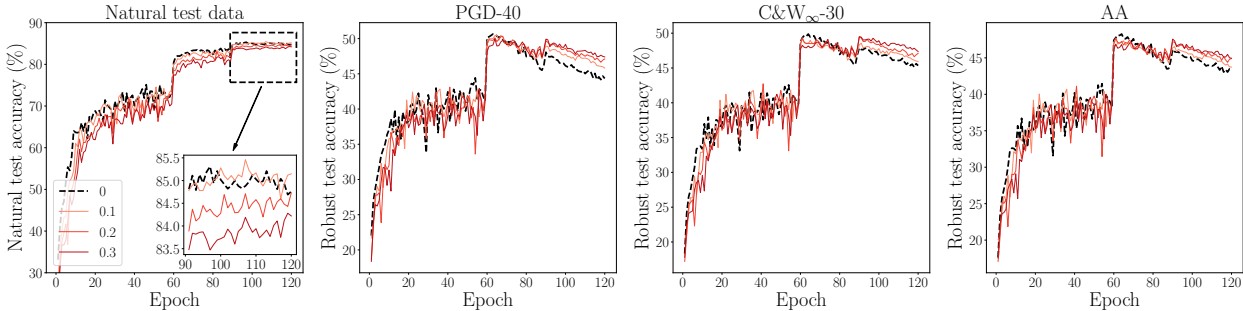

Figure 16: The learning curves of ResNet-18 trained by AT with different NL injection in inner maximization and outer minimization. The number in the legend represents noise rate $\eta$.

Figures 14 and 15 show that, with the increasing of $\eta$ (the color gradually becomes deeper), the natural test accuracy (the leftmost panels) on natural test data decreases; while the robust test accuracy (the right three panels) on adversarial data evaluated at the last checkpoint and at the best checkpoint increases simultaneously. For example, when $\eta$ equals to 0.1 (blue lines with different shades) in Figure 14, the natural test accuracy at Epoch 120 is comparable with SAT (black dashed line), and the robust test accuracy at Epoch 120 is obviously above the black dashed line. Such observation manifests that injecting NL in both inner maximization and outer minimization can alleviate robust overfitting and even improve the best-checkpoint robustness.

**Observation (iv)** Figure 16 demonstrates the learning curves of ResNet-18 trained by AT with different NL injection in inner maximization and outer minimization. In details, at every minibatch, we choose a portion of data to flip labels in inner maximization and another portion of data to flip labels in outer minimization. In Figure 16, NL is generated by symmetric flipping. The noise rate $\eta$ keeps same in both inner maximization and outer minimization, and $\eta \in \{0.1, 0.2, 0.3\}$. The detailed training settings (e.g., the optimizer and learning schedule) are the same as Section 3.1. Figure 16 shows that adversarial robustness is slightly degraded, which indicates that label-flipped adversarial data no longer serve the regularization purpose and are no longer adversarial as well to enhance robustness.

## D   Extensive Experimental Details

**Computing resources.** All experiments were conducted on a machine with Intel Xeon Gold 5218 CPU, 250GB RAM and six NVIDIA Tesla V100 SXM2 GPU, and three machines with Intel Xeon Silver 4214 CPU, 128GB RAM and four GeForce RTX 2080 Ti GPU.

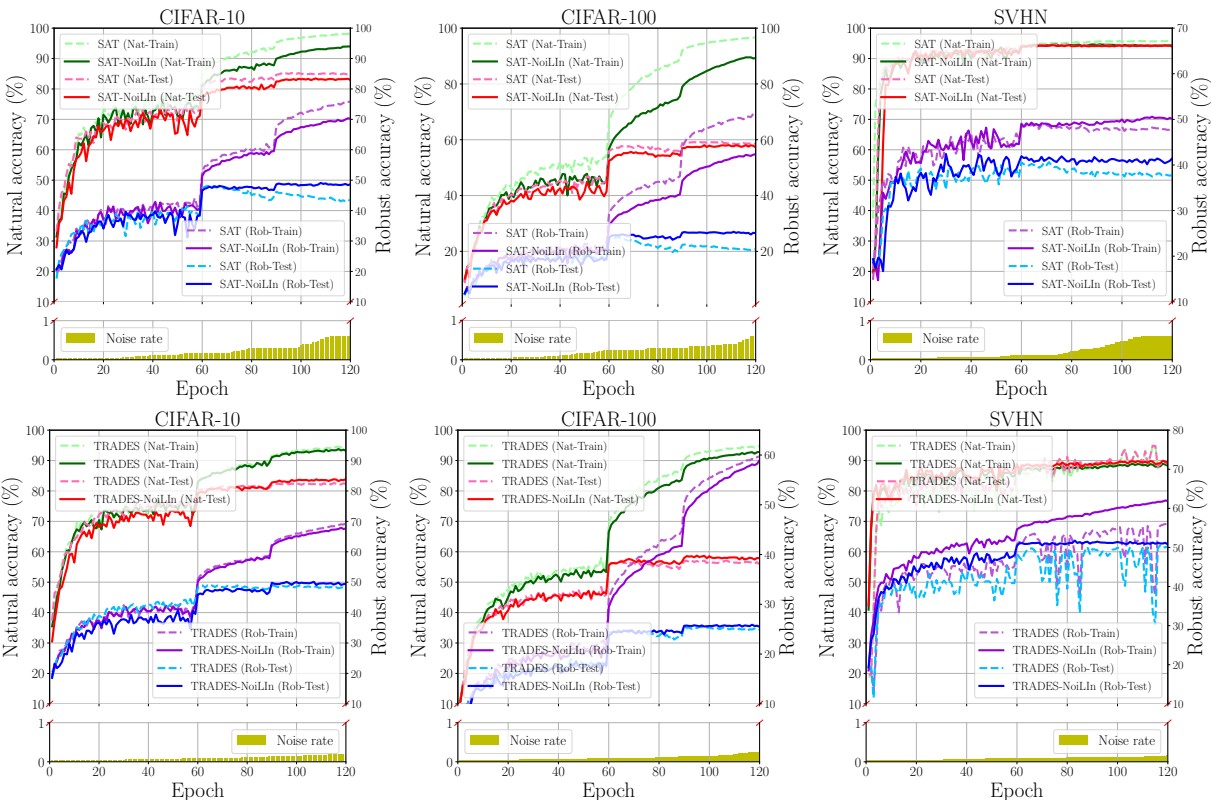

Figure 17: Evaluations on ResNet-18. We compare generalization (green/red lines for training/test data) and robustness (purple/blue lines for training/test data, evaluated by the strongest AA attack) between our NoiLIn (solid lines) and two AT methods such as SAT and TRADES (dash lines) on three datasets—CIFAR-10, CIFAR-100, and SVHN. The yellow histogram below each figure reflects the change of NL injection rate over the training process.

**Selection criterion for the best checkpoint.** We selected the best checkpoint based on robust test accuracy on AA Croce & Hein (2020) adversarial data when using ResNet-18. When using Wide ResNet, we use PGD-10 adversarial data for the consideration of speed. Note that the best checkpoints appear in Figure 6, Figure 5(b), Figure 7, and all tables.

### D.1 Relation with Label Smoothing

AT-LS Cheng et al. (2020); Pang et al. (2021) has been comprehensively investigated among different smoothing levels $\rho$. Pang et al. (2021) pointed out AT with LS only in outer minimization ("AT-LS-outer") under a mild smoothing level (e.g., $\rho = 0.1$) can slightly improve adversarial robustness. Therefore, we sampled $\rho$ from $\{0.05, 0.1, 0.2, 0.3\}$. Training details (e.g., the optimizer and learning rate schedule) of SAT, AT-LS-outer under different $\rho$, and SAT-NoiLIn (Symmetric) kept same as Section 5.1. We compared the performance of SAT, AT-LS-outer under different $\rho$, and SAT-NoiLIn (Symmetric) using ResNet-18 on CIFAR-10 dataset in Table 5. We only reported AT-LS-outer ($\rho = 0.1$) in Table 2 for the reason that AT-LS-outer ($\rho = 0.1$) achieves the most robustness against AA attacks at the best checkpoint among AT-LS-outer under different $\rho$.

**AT-NoiLIn relieves robust overfitting while AT-LS does not.** Compared with AT-LS under different $\rho$, AT-NoiLIn indeed alleviates robust overfitting. The absolute value of the robust test accuracy gap (in the parentheses at the right two columns of Table 2) obtained by AT-NoiLIn is consistently smaller than that obtained by AT-LS under different $\rho$. In addition, we observed AT-NoiLIn achieves better performance

Table 5: Comparisons between SAT, AT-LS-outer Pang et al. (2021) under different $\rho$ and SAT-NoiLIn (Symmetric). We reported the test accuracy of the best checkpoint and that of the last checkpoint as well as the gap between them—"best/last (gap)".

| Defense | Natural | C&W$_\infty$-100 | AA |
|---|---|---|---|
| SAT Madry et al. (2018) | 81.97/84.76 (+2.79) | 49.53/45.12 (−4.71) | 48.09/43.30 (−4.79) |
| AT-LS-outer ($\rho = 0.05$) | 82.38/85.19 (+2.81) | 49.60/44.95 (−4.65) | 47.86/43.01 (−4.79) |
| AT-LS-outer ($\rho = 0.1$) | 82.76/85.15 (+2.39) | 50.06/45.40 (−4.66) | 48.60/43.44 (−5.16) |
| AT-LS-outer ($\rho = 0.2$) | 82.80/85.19 (+2.39) | 49.87/45.71 (−4.16) | 48.42/43.96 (−4.46) |
| AT-LS-outer ($\rho = 0.3$) | 82.45/**85.70 (+3.25)** | 49.70/45.33 (−4.37) | 48.26/43.93 (−4.33) |
| SAT-NoiLIn (Symmetric) | **82.97**/83.86 (+0.89) | **51.21/50.36 (−0.85)** | **48.74/47.93 (−0.81)** |

of robustness evaluations bsed on C&W attacks and AA attacks at the best epoch than AT-LS under all different $\rho$.

**AT-NoiLIn is more computationally efficient than AT-LS.** We have stated that NoiLIn saves $(C-1)$ log operations for each data compared with LS. However, when it comes to empirical verification, we found AT-NoiLIn uses the comparable training time with AT-LS. Both AT-NoiLIn and AT-LS using ResNet-18 on CIFAR-10 dataset spent about 256 seconds per training epoch evaluated on a GeForce RTX 2080 Ti GPU. It is owing to that the GPU can parallelly conduct the log operation for each data. That is, for each adversarial data $\tilde{x}$, the GPU only needs to conduct one log operation on all $C$ elements of $p(\tilde{x})$ simultaneously.

But when AT-LS is conducted on the dataset which contains more classes ($C$ is larger), such as ImageNet Deng et al. (2009) dataset, the GPU may need to conduct at least more than one log operation for each data. Since the whole tensor of $p(\tilde{x})$ is too large for the limited GPU memory, $p(\tilde{x})$ has to be segmented into several small tensors that satisfying GPU memory for log operation. By contrast, AT-NoiLIn only needs to conduct one log operation on the $\tilde{y}$-th element of $p(\tilde{x})$ in this case. Therefore, NoiLIn framework is better than AT-LS from the perspective of computational efficiency.

### D.2 NoiLIn under Different Learning Rate Schedulers

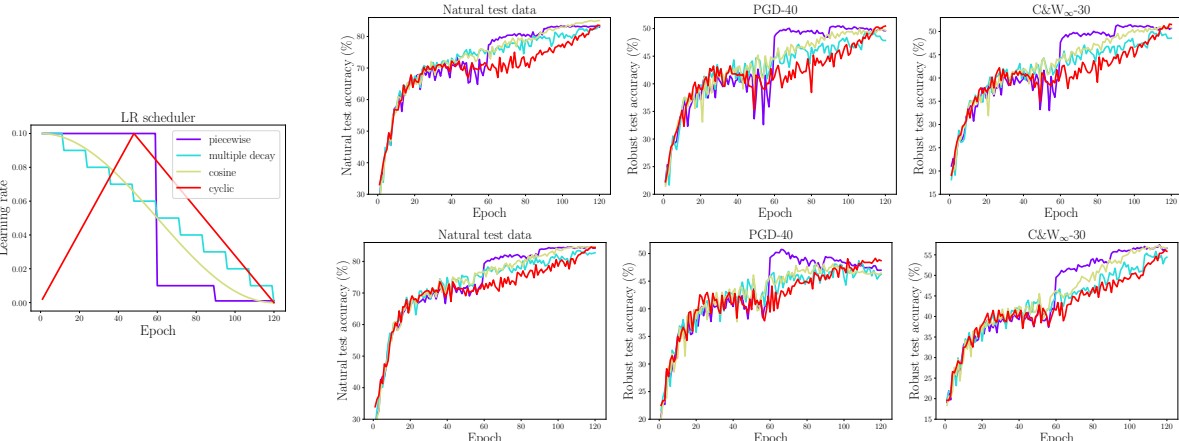

Figure 18: The leftmost panel shows the learning rate w.r.t. epoch under different LR schedulers. The right panels demonstrate the learning curves of AT-NoiLIn under various LR schedulers using symmetric-flipping NL (the upper panels) and pair-flipping NL (the lower panels), respectively.

We conducted SAT-NoiLIn using pair-flipping NL under different LR schedulers using ResNet-18 on CIFAR-10 dataset. All the training settings, such as the optimizer and hyperparameters of noise rate, kept exactly same as Section 5.1 except the LR scheduler. The learning rate w.r.t. epoch under different LR schedulers is shown in the leftmost panel of Figure 18. We demonstrated the learning curves of AT-NoiLIn (Symmetric) in

Table 6: Evaluations of SAT-NoiLIn with pair-flipping NL using ResNet-18 on CIFAR-10 dataset under various LR schedulers. We reported the test accuracy of the best checkpoint and that of last checkpoint as well as the gap between them—"best/last (gap)".

| Defense | LR scheduler | Natural | C&W$_\infty$-100 | AA |
|---|---|---|---|---|
| SAT Madry et al. (2018) | piecewise | 81.97/84.76 (+2.79) | 49.53/45.12 (−4.71) | 48.09/43.30 (−4.79) |
| SAT-NoiLIn (Pair) | piecewise | 81.29/84.26 (**+2.97**) | 51.35/**56.27** (**+4.92**) | **48.30**/46.59 (−1.71) |
| | multiple decay | 81.21/82.75 (+1.54) | 50.87/54.01 (+3.14) | 46.28/45.14 (−1.14) |
| | cosine | 81.69/84.59 (+2.90) | 53.09/56.06 (+2.97) | 46.75/45.31 (−1.44) |
| | cyclic | **84.34**/84.34 (+0.00) | **55.58**/55.58 (+0.00) | 47.75/**47.75** (**+0.00**) |

the upper row of Figure 18 and AT-NoiLIn (Pair) in the lower row of Figure 18 under various LR schedulers. Figure 18 clearly indicates NoiLIn can mitigate robust overfitting under all different LR schedulers.

Further, we reported the test accuracy of the best checkpoint and that of the last checkpoint as well as the gap between them achieved by NoiLIn using pair-flipping NL in Table 6. We observed the gap between test accuracy of the best checkpoint and that of the last checkpoint largely narrows with automatic NL injection, which validates NoiLIn can relieve overfitting.

Moreover, Table 3 and Table 6 show AT-NoiLIn under cyclic LR decay simply obtains the best checkpoint at the last epoch (the gap is exactly +0.00), which suggests AT-NoiLIn under cyclic LR scheduler can help save the time for selecting the best checkpoint. Note that for selecting the best checkpoint, it is time-consuming to gain the robust test accuracy based on AA attack for ResNet-18 and PGD-10 attack for Wide ResNet over all training epochs.

### D.3 NoiLIn with Batches that Consist of both Natural and Adversarial Examples

We evaluate the performance of ResNet-18 trained with batches composed of both natural and adversarial examples via NoiLIn (namely, SAT-NoiLIn-Mix). At each epoch, the model parameters are updated via minimizing the sum of natural data's cross-entropy loss and adversarial data's cross-entropy loss. We keep other training configurations (e.g., batch size) same as NoiLIn in Section 5. We repeat the experiments 3 times with different random seeds. In Figure 19, we show the learning curve of SAT-NoiLIn-Mix.

For one thing, Figure 19 shows that the natural test accuracy of SAT-NoiLIn-Mix is consistently above that of SAT and SAT-NoiLIn. This could be attributed to incorporating natural data into each batch, thus helping to improve natural accuracy. For another thing, we find that NoiLIn still can mitigate robust overfitting when each batch is composed of natural and adversarial data, although the robust test accuracy of SAT-NoiLIn-Mix is significantly lower than that of NoiLIn. Therefore, to achieve better robustness, we recommend NoiLIn with batches that consist of only adversarial examples.

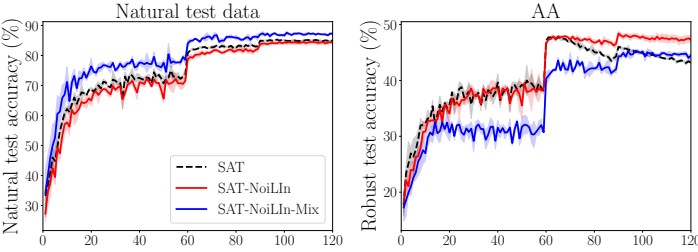

Figure 19: Comparisons between SAT, SAT-NoiLIn, SAT-NoiLIn with batches composed of both natural and adversarial examples (SAT-NoiLIn-Mix).

