# OpenReview forum: "NoiLin: Improving adversarial training and correcting stereotype of noisy labels "
_TMLR — Accepted by TMLR_

### Review · Reviewer_4KSq · 2022-04-22

**Summary Of Contributions:**

The paper studies the effect of adding label noise in adversarial training to increase the robustness of neural networks. The authors experiment with both adding noise to the inner maximization problem of adversarial training (finding an adversarial example), and the outer minimization problem (minimizing the empirical risk on the adversarial examples). Based on exploratory experiments, the authors focus on adding noise to the outer minimization problem as they find that injecting noisy labels there helps reduce adversarial overfitting. Adversarial overfitting is an effect highlighted in earlier work where the adversarial test accuracy noticeably decreases as training progresses. To address adversarial overfitting most effectively via the injection of noisy labels, the authors propose a scheme where the fraction of noisy labels in each batch is dynamically adjusted based on the amount of adversarial overfitting on a validation set.

After narrowing the scope of investigation to noisy labels in the outer minimization, the authors test their method on three datasets (CIFAR-10, CIFAR-100, and SVHN) and find a small but consistent reduction in adversarial overfitting. The authors then compare to multiple baselines and conduct a range of ablation studies to measure the empirical performance of their method in more detail.

**Broader Impact Concerns:**

The paper offers incremental improvements for a problem that has been studied extensively in the literature over the past few years (adversarial robustness on image classification datasets like CIFAR-10). Hence the paper raises no new ethical concerns and from my perspective does not require a Broader Impact Statement.

**Requested Changes:**

- As an additional baseline, it could be helpful if the authors also compare to training a network with batches that consist of both standard examples and adversarial examples.

- A summary results table similar to Table 1 but for CIFAR-100 and SVHN would make it easier to compare the effect of the proposed method to relevant baselines.

- Why do only some of the plots and tables show both the "symmetric" and "pair" version of NoiLIn?

- It would be good to know if the method also improves when additional training data is used as extra unlabeled data has been shown to increase adversarial robustness. The following paper provides a good overview of the currently best methods in terms of adversarial robustness: https://arxiv.org/abs/2010.03593  Ideally the advantages of NoiLIn would be additive with the currently best variants of adversarial training and yield a new state-of-the-art.


Small changes:

- "In Table , we also" - table number missing near the bottom of Page 8.

- Typo "abalation study" at the beginning of Section 5.3.

- Figure 5 b: The y-axis label on the left plot should be "Natural test accuracy (%)".

- The difference between the abbreviations "AT" and "SAT" is unclear.

- The third sentence near the top of Page 11 seems to have a spurious "kept".


**Strengths And Weaknesses:**

Strengths:
- The proposed method consistently reduces adversarial overfitting.

- The evaluation of adversarial robustness uses multiple attacks.

- The authors conduct a range of ablation studies to empirically understand the behavior of their method.

- The paper is well written.


Weaknesses:
- Overall the gains of the proposed method are modest. Compared to relevant baselines on CIFAR-10, the additional robustness is around 1 percentage point and the increase in standard (non-adversarial) accuracy of the same model is about 2 percentage points.

- It is unclear if the proposed method also helps on larger datasets such as ImageNet. However I understand adversarial training on ImageNet requires substantially more compute and may not be feasible for the authors.

---

> ### Author Response · Authors · 2022-05-09
> **Response to Reviewer 4kSq.**
>
> Many thanks for the detailed comments. We have uploaded a revised version with added experiments and corrected typos. The major changes are highlighted in blue.
>
> **Q**: As an additional baseline, training a network with batches that consist of both standard examples and adversarial examples.
>
> **A**: In appendix D.4, we conduct additional experiments that evaluate the performance of ResNet-18 trained with each batch composed of both natural and adversarial examples by the NoiLIn method (namely, SAT-NoiLIn-Mix). The results (Fig. 19 in Appendix D.4) show that incorporating the natural data leads to better standard accuracy but worse robustness, which has similar observations to the ``NL injection in inner maximization”.
>
> **Q**: "symmetric" and "pair" versions of NoiLIn.
>
> **A**: In the main paper, we mainly show noisy label (NL) injection via the symmetric way where labels are flipped to other classes at random with a uniform distribution. However, the NL could also be injected in a different way such as the pair-flipping way where labels are flipped between adjacent classes that are prone to be mislabeled. The experiments of pair-flipping are mostly put into the appendix.
>
> **Q**: Leveraging additional unlabeled data.
>
> **A**: At the end of Section5.3, we conduct and add additional experiments that leverage additional unlabeled (U) data by Carmon et al 2019. We train WRN-28-10 with the cosine decay learning rate schedule. We set the ratio of labeled-to-unlabelled data per batch to 3:7 following Gowel et al. 2021. The total training epoch is 200 and the batch size is 256. Other hyper-parameters for injecting label noise are the same as Section 5. We compare the natural test accuracy and robust test accuracy (evaluated with AA attack) between NoiLIn and Robust Self-training (Carmon et al. (2019)). Note that the results of Robust Self-training with extra data (Carmon et al. (2019)) are copied from the AutoAttack leaderboard by Croce et al. (2022). Table 4 in the revised paper (The copied result as follows) shows that NoiLIn can enhance both natural and robust test accuracy, which validates the superiority of the NoiLIn method. We guess that the reason is that NoiLIn naturally performs well under a noisy-training set (that corroborates “NoiLIn under noisy-label training set” in section 5.3) since the pseudo-labels on the additional unlabeled data are noisy.
>
> | Method | standard accuracy | robust accuracy (by AA attack) |
> | --------------- | --------------- | --------------- |
> | Robust Self-training (Carmon et al. (2019)) |  89.69 | 59.53 |
> | NoiLIn with additional unlabeled data | **91.11** | **59.91**|
>
>
> The current state-of-the-art results are due to super-large models (such as WRN-70-16) and large data, which are computationally prohibitive from our end.

---

### Review · Reviewer_TrgE · 2022-04-23

**Summary Of Contributions:**

The paper proposes a new empirical method NoiLIn to improve robustness in adversarial training: inject noisy labels into the adversarial training procedure's outer minimization, and the level of noise introduced in each epoch gets increased when robust overfitting happens. The method is motivated by their empirical observation that having noisy labels in the outer minimization helps reduce robust overfitting. However, they also observed that injecting too much noise destabilizes the training, which I believe is the main reason that NoiLIn gradually increases the level of noise along with training epoch. The experiment section shows that NoiLIn improves SOTA results, and comparison with label smoothing method is provided.

**Broader Impact Concerns:**

There are no concerns on the ethical implications. Defending adversarial attack is important to safely applying machine learning techniques.

**Requested Changes:**

Adding the related paper and explaining that $\textbf{the label noise has been used in a positive way for adversarial training}$ is critical.

All the proposed adjustments below would strengthen the work:

* a few grammatical errors (note that this list is not holistic since the reviewer was not reading word by word):
  * P1: "in labels for the learning the neighborhoods"
  * P1: "from a unknown distribution"
  * P7: "NL injection in data at has similar effect"
  * P9: "The training settings of ... keeps "
  * P9: "NoiLIn have ..."
* some abbreviations appear before their explanations. For example, NoiLIn (Pair) appears in Figure 5, but was first explained in Figure 6's caption.

* More mathematical rigour would be appreciated: for example, when mentioning "labels $y$ are flipped at random with the uniform distribution", does the support set include the true label or not? As another example, condition for determine robust overfitting is stated as "$\sum_{i=e-\tau}^{e}\mathcal{A}\_i<\sum_{j=e-\tau-1}^{e-1}\mathcal{A}\_j$", why not just say $\mathcal{A}\_e < \mathcal{A}\_{e-\tau-1}$?

* Other writing related (suggestions):
  * avoid using "etc" in academic paper writing.
  * The second and third paragraph under Observation (i): it is better to illustrate the effects using Figure 2(b) first. Explaining Figure 2(b) in the third paragraph seems redundant/in the wrong order because one may already understand the key points through the second paragraph.

**Strengths And Weaknesses:**

# Strengths:
* The paper includes solid experimental results. The observations related to noisy label injection in inner maximization / outer minimization are accompanied with intuitive explanations that are easy to follow.
* The proposed algorithm is a "meta" algorithm that can be combined with existing algorithms due to the fact that state-of-the-art algorithms are not exploiting noisy labels.

# Weaknesses
* Writing requires more proofread and more mathematical rigour. Also, section 3.3 seems somewhat redundant.
* Missing literature: The fact that more than half page discussing NoiLIn's relation with label smoothing while the original label smoothing paper is not cited is a bit concerning to me.
* Correcting stereotype of noisy labels in the title is a bit misleading here: At least the label smoothing paper (https://arxiv.org/pdf/1910.11585.pdf) proposes using label noise as regularization to achieve adversarial robustness, which is intuitively very natural. Therefore, the novelty of the paper given that "there may be no stereotype in using label noise for adversarial training" is relatively small.

---

> ### Author Response · Authors · 2022-05-09
> **Response to Reviewer TrgE.**
>
> Many thanks for the detailed comments. We have uploaded a revised version with the improved citations, discussions and corrected typos. The major changes are highlighted in blue.
>
> **Q** Adding the related paper and explaining that the label noise has been used in a positive way.
>
> **A** There are debates on whether label smoothing and logit squeezing [1], and logit pairing [2] genuinely benefit adversarial robustness or mask gradients for overly reporting robustness. This paper avoids such a debate by evaluating our NoiLIn method using the strongest AutoAttack (AA, which should thwart the concerns of gradient obfuscations [3]. We have included the citations and discussions in the part of the related work in revision.
>
> [1] Label smoothing and logit squeezing: a replacement for adversarial training?\
> [2] Logit pairing methods can fool gradient-based attacks\
> [3] Obfuscated Gradients Give a False Sense of Security: Circumventing Defenses to Adversarial Examples in ICML18
>
>
> **Q**  Labels y are flipped at random with the uniform distribution.
>
> **A** The support set includes the true label.
>
> **Q**  Condition for determining robust overfitting.
>
> **A**  We want to exclude the accuracy fluctuations over epochs that lead to a misjudgment of robust overfitting.

---

### Review · Reviewer_8SzM · 2022-04-26

**Summary Of Contributions:**

The paper discusses the use of noisy labels in adversarial training. Noisy label can be introduced in the outer or the inner optimisation problem of PGD-type training. The paper discusses both of them and how they impact clean and adversarial accuracy. Further, the paper proposes an algorithm to introduce even more noise in adversarial training as a method to further improve adversarial accuracy by preventing robust overfitting.

**Broader Impact Concerns:**

There are not broader ethical or impact concerns.

**Requested Changes:**

* Please repeat the experiments to show **confidence intervals** for all results.
* Figure 5(b) left's y axis label is probably incorrect. Shouldn't it be  _Natural test accuracy_ ?
* Please discuss the importance of Figure 3(b) --- why is the "diversity" as defined in the paper useful and how does it add to the contribution of the paper ?
* Please look at modifying the introduction especially the part that discusses the role label noise plays in adversarial training. I find the current explanation that label noise somehow allows for overlap between different classes to not have a detrimental effect during AT to be unsatisfactory as it is not supported by any experiments. Rather I find the discussion in Donhauser et. al. that label noise is a natural regulariser as it avoids the max $\ell_2$ margin interpolator to be more reasonable. Could I please ask the authors to discuss this ?

**Strengths And Weaknesses:**

Some of the particularly good points about the paper, in my opinion, are as follows

* I found the discussion of the role of the noise in the outer and inner maximisation problem interesting, intuitive, and borne out by experiments.
* I think the authors have made a very good attempt at discussing/mentioning relevant existing literature.
* The visualisations and the notations are also clear and helpful. I appreciated showing magnified views of the training curves as this shows, in greater detail, the most relevant part of the training.
* I think the experimental evaluation is also sufficient. CIFAR10, CIFAR100, and SVHN are three commonly used datasets in adversarial training literature and the authors have used all of them. The paper uses AutoAttack to measure robustness, which is now a standard and a powerful method.


Weakness
* The experimental advantage of NoiLin is minor and insignificant. It is minor because the difference from not using NoiLin is often less than 1% and insignificant because it does not have any kind of error bars. For example in table 1, for AutoAttack NoiLin is always within 1% of its competitor. Similar results are seen in its comparison with label smoothing.  Therefore, I am not convinced that NoiLin has any actual advantage.
* I found the discussion of the importance of noisy labels in the introduction very far-fetched and unsupported by experimental evidence. In essence, the main argument the paper makes is that "noisy labels are helpful for AT" as it allows for AT to account for the overlap in the input domain between different classes. However, there is no experimental evidence to show i) that this overlap exists in noiseless datasets ii) NoiLin actually helps avoid robust overfitting by this exact overfitting. Therefore, this discussion in the introduction seems to unrelated to the rest of the paper and potentially incorrect.
* I am confused about the implication of Figure 3(b).  Why is having a larger variance in the loss of training data points more helpful and indicative of _helpful_ diversity ? Maybe I am missing the argument here and would encourage the authors or clarify the importance of this figure.
* Section 4.4 and Figure 6 in Donhauser et. al have a very similar observation that adding noisy labels help avoid robust overfitting for adversarial training and they do this by avoiding the $\ell_2$-max margin interpolator. How is the conclusion of this paper different from theirs and, according to the authors, is the underlying phenomenon the same in both cases ?



Donhauser, Konstantin, et al. "Interpolation can hurt robust generalization even when there is no noise." Advances in Neural Information Processing Systems 34 (2021).

---

> ### Author Response · Authors · 2022-05-09
> **Response to Reviewer 8SzM**
>
> Many thanks for the detailed comments. We have uploaded a revised version with the improved experiments (by adding confidence intervals) and improved discussions. The major changes are highlighted in blue.
>
> **Q** Add confidence intervals in experimental results.
>
> **A** We have re-run all experiments using three different random seeds in Figure 4 and Table 1 (The confidence interval of Table 1 is put in Table 7 of appendix D3).
> NoiLIn adds NL in the curriculum learning way, which will not destabilize the training but surprisingly enhance the training stability (i.e., NoiLIn's confidence interval is smaller).
> Therefore, we can say the improvements by NoiLIn are significant!
>
> **Q** Overlaps in a noiseless dataset in AT.
>
> **A** [1] has shown even the benchmarked dataset contains the noisy labels in the training set.
> [2] has shown that class overlaps may not happen in the input layers but in the intermediate layers of DNNs, and AT has severe cross-over mixture problems between different classes.
> Although there are some claims that there are no overlaps between two classes in the real-world dataset, empirical results of AT always show a severe tradeoff between accuracy and robustness [3]. \
> [1] How benign is benign overfitting? In ICLR21\
> [2] Attacks which do not kill training make adversarial learning stronger. In ICML20\
> [3] Robustness May Be at Odds with Accuracy in ICLR19
>
> **Q** Discussing the importance of Figure 3(b) --- why the "diversity" as defined in the paper is useful
>
> **A** The diversity defined in this paper is “how differently a model sees data over the learning process’’. Let us make an analogy of comparing training a model with educating a kid over the years. If this kid continuously learns the same things, this kid will be overfitted and brainwashed.
> Instead, if this kid learns things differently and even from occasional errors, this kid will be more resilient and robust. Similarly, we encourage the model to learn differently over training epochs and even from mistakes, making the model less overfitted and robust.
>
> **Q** The discussion of Donhauser et. al. [4] that label noise is a natural regulariser
>
> **A** [4] theoretically focused on robustly separable data, studied the robust generalization performance of regularized estimators, and highlighted the importance of regularization in improving the robust risk.
> [4] found that some noisy labels in the training set serve as a regularizer that can benefit the adversarial robustness.
> However, [4]'s setting is very different from ours. We treat the original data noiseless and inject differently noisy labels at each training epoch. We empirically verify noisy labels (NLs) injections relieve the robust overfitting. Indeed, we can also treat NLs injection as a regularization that shares the similar arguments to [5].
>
> [4] Interpolation can hurt robust generalization even when there is no noise, in NeurIPS 2021\
> [5] Disturblabel: Regularizing CNN on the loss layer, in CVPR 2016

---

> > ### Comment · Reviewer_8SzM · 2022-05-22
> > **Reponse and further improvement**
> >
> > * I would stress that the authors try to integrate the standard deviations in Table 1 (at least in the main text) as opposed to putting them in the appendix as these are really important.
> >
> > * I am still not convinced by the argument for "overlaps in noiseless dataset in AT". Perhaps, the authors can edit the paper to make this argument more rigorously.
> >
> > * Regarding the discussion in Figure 3(b), the discussion that if the "kid learns things differently and even from occasional errors, this kid will be more resilient and robust", is too generic and frankly a very non-rigorous and unfalsifiable response. It does not help at all in my understanding of the questions.
> >
> > * Could you clarify what you mean by the setting is very different ? As you say [4] deals with robustly separable data, then noiseless data satisfies that criterion. Could you please be more precise regarding how you mean they are different ?

---

> > > ### Author Response · Authors · 2022-05-26
> > > **The responses to Reviewer 8SzM**
> > >
> > > Many thanks for your replies. We have made further revisions according to your comments.
> > >
> > > **Q**  Integrate the standard deviations in Table 1 (at least in the main text) as opposed to putting them in the appendix \
> > > **A** In the revised version, we have integrated the standard deviations into Table 1.
> > >
> > >
> > > **Q**  Overlaps in a noiseless dataset in AT \
> > > **A** We have revised the introduction and abstract (highlighted in blue) and weakened the argument by “stating several facts that motivate our study on injecting noisy labels in AT.”
> > > The revision is also copied as follows.
> > >
> > > AT empirically shows a severe tradeoff between the natural accuracy of natural test data and the robust accuracy of adversarial test data.
> > > Besides, Zhang et al. showed AT has a cross-over mixture problem, where the adversarial variants of the natural data cross over the decision boundary and fall in the other-class areas.
> > > Even, Sanyal et al. showed the benchmark dataset (such as CIFAR) contains some noisy data points. Furthermore. Donhauser et al. showed an unorthodox way to yield an estimator with a smaller robust risk, i.e., flipping the labels of a fixed fraction of the training data. The above facts motivate us to explore manipulating labels for benefits that were largely overlooked by the existing AT studies.
> > >
> > > **Q** Figure 3 (b)  \
> > > **A** Sorry that the analogy (that never appears in the submission) does not help the understanding.
> > > I have revised the paper and weakened the argument to that “Figure 3 (b) echoes the empirical observations in Figure 3 (a), which implies that the high data diversity may impede the undesirable issue of robust overfitting. The changes are highlighted in blue.
> > >
> > >
> > > **Q** Difference between Donhauser et al.’s setting and NoiLIn. \
> > > **A** Sorry for my confusion. Indeed, both papers assume the clean training set.
> > > Donhauser injects noisy labels into the training set, and then the noisy labels are fixed over the training process. Therefore, a specific data point always sees the correct or flipped label over the entire training process.
> > > Differently, NoiLIn flips labels at each training epoch. Therefore, this specific data point often sees the correct label and occasionally sees the flipped label over the entire training process.

---

### Decision · Action_Editors · 2022-06-09

**Recommendation:** Accept as is

**Comment:**

This paper proposed a noisy label injection method to improve adversarial training. All reviewers acknowledged the merits of this work and unanimously recommend acceptance. In the rebuttal phase, there were several issues regarding the explanations and intuitions of the proposed method, as well as some ambiguous descriptions of the presented results. In the revised version, the authors have successfully addressed all reviewers' concerns. Although the performance improvement over standard adversarial training can sometimes be marginal, the improvement is consistently observed on all datasets. Therefore, the technical claims are solid. I recommend accepting the submission as is.